# Engineering antiviral immune-like systems for autonomous virus detection and inhibition in mice

Yidan Wang [1,2], Ying Xu[1], Chee Wah Tan [3], Longliang Qiao[1], Wan Ni Chia [3], Hongyi Zhang[4], Qin Huang[1], Zhenqiang Deng[1], Ziwei Wang[1], Xi Wang[5,6], Xurui Shen[5,6], Canyu Liu[6,7], Rongjuan Pei[7], Yuanxiao Liu[1], Shuai Xue [1,8], Deqiang Kong [1], Danielle E. Anderson [3], Fengfeng Cai [4], Peng Zhou[5], Lin-Fa Wang [3,9] ✉ & Haifeng Ye [1] ✉

The ongoing COVID-19 pandemic has demonstrated that viral diseases represent an enormous public health and economic threat to mankind and that individuals with compromised immune systems are at greater risk of complications and death from viral diseases. The development of broad-spectrum antivirals is an important part of pandemic preparedness. Here, we have engineer a series of designer cells which we term autonomous, intelligent, virus-inducible immune-like (ALICE) cells as sense-and-destroy antiviral system. After developing a destabilized STING-based sensor to detect viruses from seven different genera, we have used a synthetic signal transduction system to link viral detection to the expression of multiple antiviral effector molecules, including antiviral cytokines, a CRISPR-Cas9 module for viral degradation and the secretion of a neutralizing antibody. We perform a proof-of-concept study using multiple iterations of our ALICE system in vitro, followed by in vivo functionality testing in mice. We show that dual output $ALICE_{SaCas9+Ab}$ system delivered by an AAV-vector inhibited viral infection in herpetic simplex keratitis (HSK) mouse model. Our work demonstrates that viral detection and antiviral countermeasures can be paired for intelligent sense-and-destroy applications as a flexible and innovative method against virus infection.

With global climate change, we will confront more emerging and reemerging human diseases[1] and the lack of effective antivirals is taking a toll on human lives and social wealth[2]. As part of pandemic preparedness, there is an urgent need to establish effective antiviral strategies against "Disease X"[2]. The human innate immune response is the first line of defense against viral pathogens[3], but there are vast differences in the efficacy of innate immune functions between individuals, especially immunocompromised individuals who are

[1]Shanghai Frontiers Science Center of Genome Editing and Cell Therapy, Biomedical Synthetic Biology Research Centre, Shanghai Key Laboratory of Regulatory Biology, Institute of Biomedical Sciences and School of Life Sciences, East China Normal University, Dongchuan Road 500, Shanghai 200241, China. [2]Chongqing Key Laboratory of Precision Optics, Chongqing Institute of East China Normal University, Chongqing 401120, China. [3]Programme in Emerging Infectious Diseases, Duke-NUS Medical School, Singapore, Singapore. [4]Department of Breast Surgery, Yangpu Hospital, School of Medicine, Tongji University, 450 Tengyue Road, Shanghai 200090, China. [5]CAS Key Laboratory of Special Pathogens, Wuhan Institute of Virology, Chinese Academy of Sciences, Wuhan 430071 Hubei, China. [6]University of Chinese Academy of Sciences, Beijing, China. [7]State Key Laboratory of Virology, Wuhan Institute of Virology, Chinese Academy of Sciences, Wuhan 430071 Hubei, China. [8]Department of Biosystems Science and Engineering, ETH Zurich, CH-4058 Basel, Switzerland. [9]SingHealth Duke-NUS Global Health Institute, Singapore, Singapore. ✉e-mail: linfa.wang@duke-nus.edu.sg; hfye@bio.ecnu.edu.cn

2–30 times more likely to contract cytomegalovirus (CMV), hepatitis B virus (HBV), and herpes simplex virus types 1 and 2 (HSV-1 and HSV-2) infection[4]. HSV-1 infection can cause a wide variety of diseases, including herpetic simplex keratitis (HSK), which has a high mortality rate if untreated[5]. Generally, HSK is caused by HSV-1 infection in the cornea, which is a leading cause of blindness and viral encephalitis in the developed world[6]. Early recognition of viruses is initiated by STING [stimulator of interferon (IFN) genes] in immune cells. STING recognizes double-stranded DNA or RNA released from an invading virus[7,8] and results in the translocation of transcription factor IRF3 and the production of chemokines and proinflammatory cytokines that recruit phagocytes to the site of infection to eliminate virus in mammalian cells[9].

Synthetic biology-based molecular diagnostics have been created by engineering natural biological components for practical applications[10–12]. Programmable RNA sensors called toehold switches were designed to bind and sense virtually any RNA sequence and can be used to detect viruses[13]. The freeze-dried, paper-based, cell-free protein expression platform allows for the deployment of the toehold switch sensor for virus detection outside of a research laboratory anytime, anywhere, and at room temperature[14,15].

The ongoing biological revolution stemming from the discovery of clustered regularly interspaced short palindromic repeats (CRISPR) has led to the development of nucleic acid-based pathogen detection technologies, notably the CRISPR-based diagnostic tools [SHERLOCK[16,17], DETECTR[18], and HOLMES[19]], which were based on the collateral effect of an RNA-guided and RNA-targeting CRISPR effector Cas13 or Cas12a. These technologies are of high sensitivity and specificity in the detection of target-specific virus[20].

In addition to these CRISPR-based detection technologies, CRISPR-associated protein 9 (Cas9)-nuclease-based technologies have been developed to degrade virus genomic materials with demonstrated in vitro potential with SARS-CoV-2[21], influenza[21], human immunodeficiency virus (HIV)[22], HBV[23], human papillomavirus (HPV)[24], and HSV-1[25]. However, the delivery of Cas9 using a lentiviral- or adeno-associated virus (AAV)-vector driven by constitutive promoters can lead to sustained Cas9 expression, anti-Cas9 immune responses, and off-target editing, which have halted the use of these technologies as antiviral therapies in clinical applications[26]. It is worth to note that the constitutive expression of Cas9 can increase the risk of the emergence of CRISPR/Cas9-resistant escape mutant virus strains[25].

Recent advances in both our understanding of immunology and insights from synthetic biology about cellular engineering have led to the technological development of "immune-like designer cells", including a self-regulated designer cell that can prevent methicillin-resistant *Staphylococcus aureus* infection[27]. Thus, there is hope for offering better clinical care to immunocompromised patients based on the development and deployment of smart cells that can compensate for inoperative or weak immune responses by acting as autonomous and potentially broad-spectrum antiviral agents.

Our research focuses on engineering antiviral immune-like systems for biomedical applications, and we envisioned that it is possible to combine viral detection and antiviral function in the same engineered cell, potentially with virus infection-triggered conditional expression of antiviral cytokines (IFN-α and IFN-β), the Cas9/sgRNA (small guide RNA) complex and even broadly-neutralizing antibodies. Such engineered intelligent "sense-and-destroy" immune-like cells should be able to autonomously detect the presence of a viral pathogen and respond with targeted antivirals. Here we report the development and deployment of autonomous, intelligent, virus-inducible immune-like (ALICE) systems. Functional ALICE cells include a closed-loop gene network comprising a destabilized version of the STING immune signal pathway sensor for the detection of viruses that can activate STING-based signaling, a synthetic virus-inducible promoter to drive the expression of antiviral cytokines (ALICE$_{im}$), and orthogonal

mechanisms of antiviral activity including a Cas9-based degradation module (ALICE$_{Cas9}$) and a neutralizing antibody (Ab) module (ALICE$_{Ab}$). We demonstrate that the ALICE system can monitor the cellular environment for the presence of viral pathogens and show that ALICE cells selectively induce antiviral responses upon the detection of a virus. ALICE$_{im}$ has the ability to enhance the host immune response and relative broad-spectrum antiviral activity due to the inducible expression of antiviral cytokines. Following proof-of-concept experiments in vitro using multiple variations of the basic ALICE system, we applied our technology in vivo in mouse models and show that the ALICE systems efficiently blocked HSV-1 replication and spread in mice at different stages of viral infection. Additionally, the dual Cas9 and neutralizing antibody system (ALICE$_{SaCas9+Ab}$) delivered by an AAV-vector could eliminate viruses via retrograde transport from corneas to trigeminal ganglia (TG) in HSK mice. Taken together, the ALICE systems might have the potential to combat refractory virus infectious diseases.

## Results

### Design and validation of an autonomous, intelligent, virus-inducible immune-like sensor (ALICE$_{sen}$)

The central component of our antiviral platform requires a sensitive sensor to detect viral nucleic acid. As a first trial, we designed an autonomous, intelligent, virus-inducible immune-like sensor (ALICE$_{sen}$) in which the destabilized STING protein is ectopically expressed in mammalian cells to function as the initial sensor molecule to detect the presence of intracellular exogenous dsDNA or RNA. The activation of destabilized STING then recruits TANK-binding kinase 1 (TBK1) and traffics from the endoplasmic reticulum to a perinuclear endosomal compartment, leading to the phosphorylation and dimerization of IRF3. Subsequently, the phosphorylated and dimerized IRF3 translocates into the nucleus and binds to synthetic promoters (P$_{ALICEx}$) positioned at genes of interest (GOI), thus enabling exogenous-viral nucleic acid-triggered transcriptional activation in mammalian cells (Fig. 1a).

To optimize a virus-responsive ALICE$_{sen}$ iteration that achieves minimal basal transgene expression in the absence of virus and maximal induction ratio in the presence of virus, we used HSV-1 as a model virus, and initially utilized stable STING. Aligned with published data[28], our immunoblotting assays of endogenous human proteins revealed no STING and cGAS expression in HEK-293T cells (Supplementary Fig. 1a). When exposing different amounts of stable STING, the maximal induction was (3.3-fold ± 0.26) (Supplementary Fig. 1b). To increase induction fold with reduced background, we fused a proteolytic tag (PEST) onto the C-terminus of the STING protein under the control of various promoters (Supplementary Fig. 1c). By promoting the proteolytic degradation of the STING protein, induction was increased to ~20-fold. We assessed the HSV-1-responsive ALICE$_{sen}$ guided by the synthetic IRF3-specific or non-specific promoters (Supplementary Fig. 1d) in HEK-293T cells and found the combination of P$_{hCMV}$−driven STING-PEST and P$_{ALICE6}$, labeled as ALICE$_{sen}$, showed the highest induction (~23-fold) of the reporter gene secreted alkaline phosphatase (SEAP) in the presence of HSV-1.

Importantly, ALICE$_{sen}$ could also be activated by infection with STING-dependent pan-genus viruses[29–32], including DENV-2, SARS-CoV-2, hCoV-229E, hepatitis C virus (HCV), HBV, adenovirus (ADV) and HSV-1 (Fig. 1b), indicating that ALICE$_{sen}$ might be used as a relative broad-spectrum sensor to detect STING-dependent viruses. Meanwhile, infection by other STING-suppression viruses[32–37] [e.g., influenza virus (H1N1), enterovirus 71 (EV-A71), pteropine reovirus-2 (PRV2P), vesicular stomatitis virus (VSV), lentivirus or adeno-associated virus (AAV)] failed to do so (Fig. 1b). ALICE$_{sen}$ has the potential to aid in the understanding of the structural and mechanistic STING pathway against virus infection. The maximal induction (~412-fold) of ALICE$_{sen}$ was reached with an enhanced green fluorescent protein

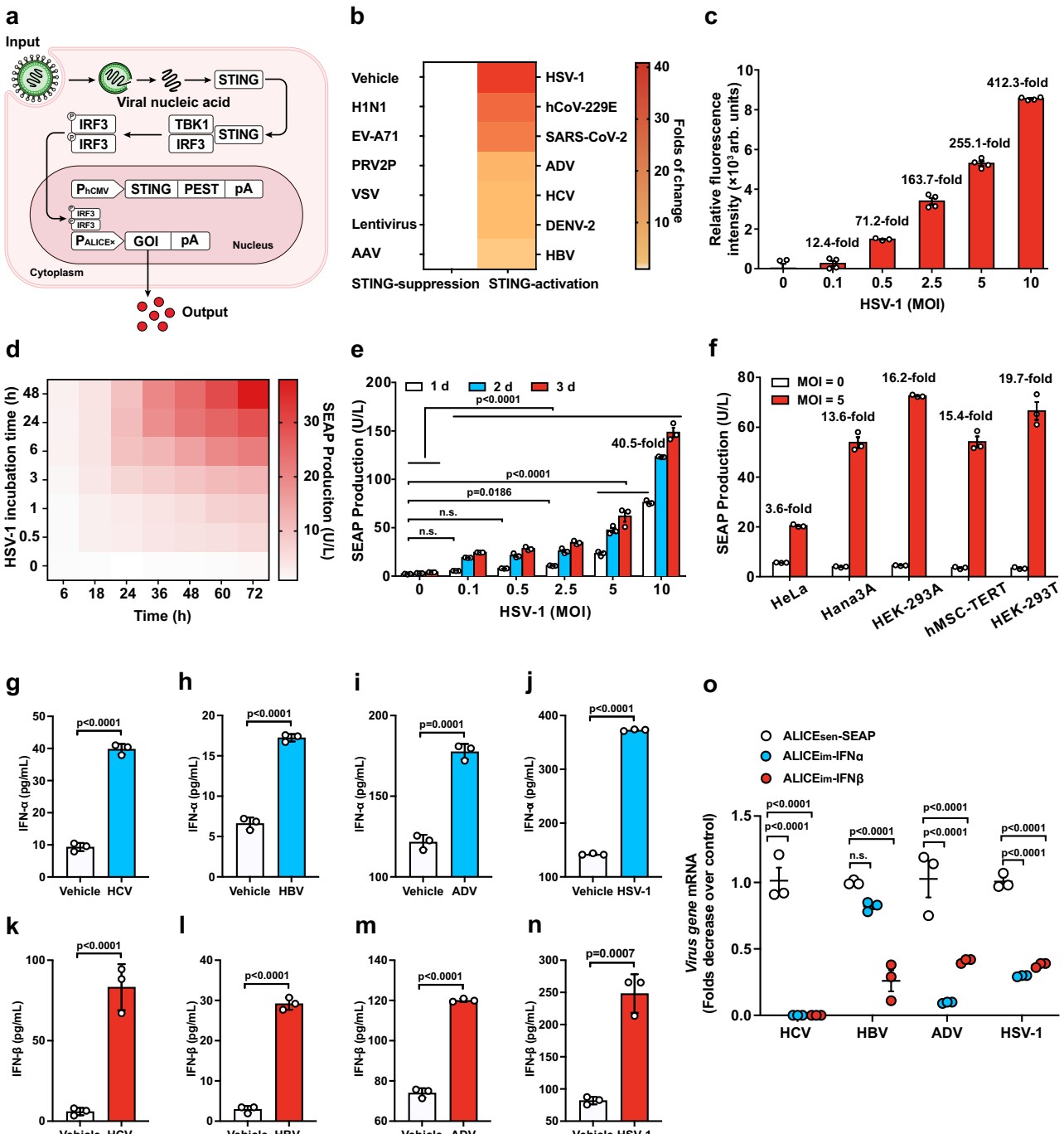

**Fig. 1 | Design and characterization of an autonomous, intelligent, virus-inducible immune-like sensor (ALICE_sen). a** Schematic illustration of the design principle for a virus sensor (ALICE_sen). Virus infects mammalian cells and releases its nucleic acid into the cytoplasm, which activates the destabilized STING (pYW274). Activated STING triggers a synthetic immune signaling pathway mediated by endogenous tank-binding kinase 1 (TBK1), resulting in activating phosphorylation and dimerization of endogenous interferon regulatory factor 3 (IRF3), which translocates into the nucleus as a dimer and initiates the expression of a given gene of interest (GOI) under the control of a "virus-inducible promotor (P_ALICEx)" sequence. **b** Fold change of SEAP in ALICE_sen induced by different STING-suppression/activation viruses. pYW274/pWS67-transgenic cells were incubated with different viruses as indicated, and SEAP in supernatant was quantified at 2 and 4 dpi (day post-infection). **c** HSV-1-dependent EGFP expression in ALICE_sen. pYW274/pYW379-transgenic HEK-293T cells were incubated with different titers of HSV-1 (MOI = 0–10) and quantified at 2 dpi. **d** Time-dependent SEAP production kinetics

of ALICE_sen. HEK_ALICE-SEAP stable cell lines were incubated with HSV-1 (MOI = 0.5) for different incubation times. SEAP levels in culture supernatants were quantified at the indicated time points. **e** HSV-1-dependent SEAP production kinetics of ALICE_sen. pYW274/pWS67-transgenic HEK-293T cells were incubated with different titers of HSV-1. White bars (1 day), blue bars (2 day), red bar (3 day). *P* values for all other groups versus HSV-1 (MOI = 0) group on the same day. **f** HSV-1-inducible SEAP production in various mammalian cell lines. **g**–**n** Virus-inducible IFN-α/IFN-β production in ALICE_im. *P* values for virus group versus Vehicle group. **o** qPCR analysis of viral genes in ALICE_im. White circles (ALICE_sen-SEAP), blue circles (ALICE_im-IFNα), red circles (ALICE_im-IFNβ). *P* values for all other groups versus ALICE_sen-SEAP group in the same virus. Data in b-o are expressed as means ± SD; *n* = 3 independent experiments in **b**, **d**–**o**; *n* = 3 or 4 independent experiments in (**c**); *P* values in **e**, **o** were calculated by two-way ANOVA with Bonferroni's post hoc test; *P* values in **g**–**n** were calculated by two-tailed unpaired *t*-test; n.s. not significant. Source data are provided as a Source Data file.

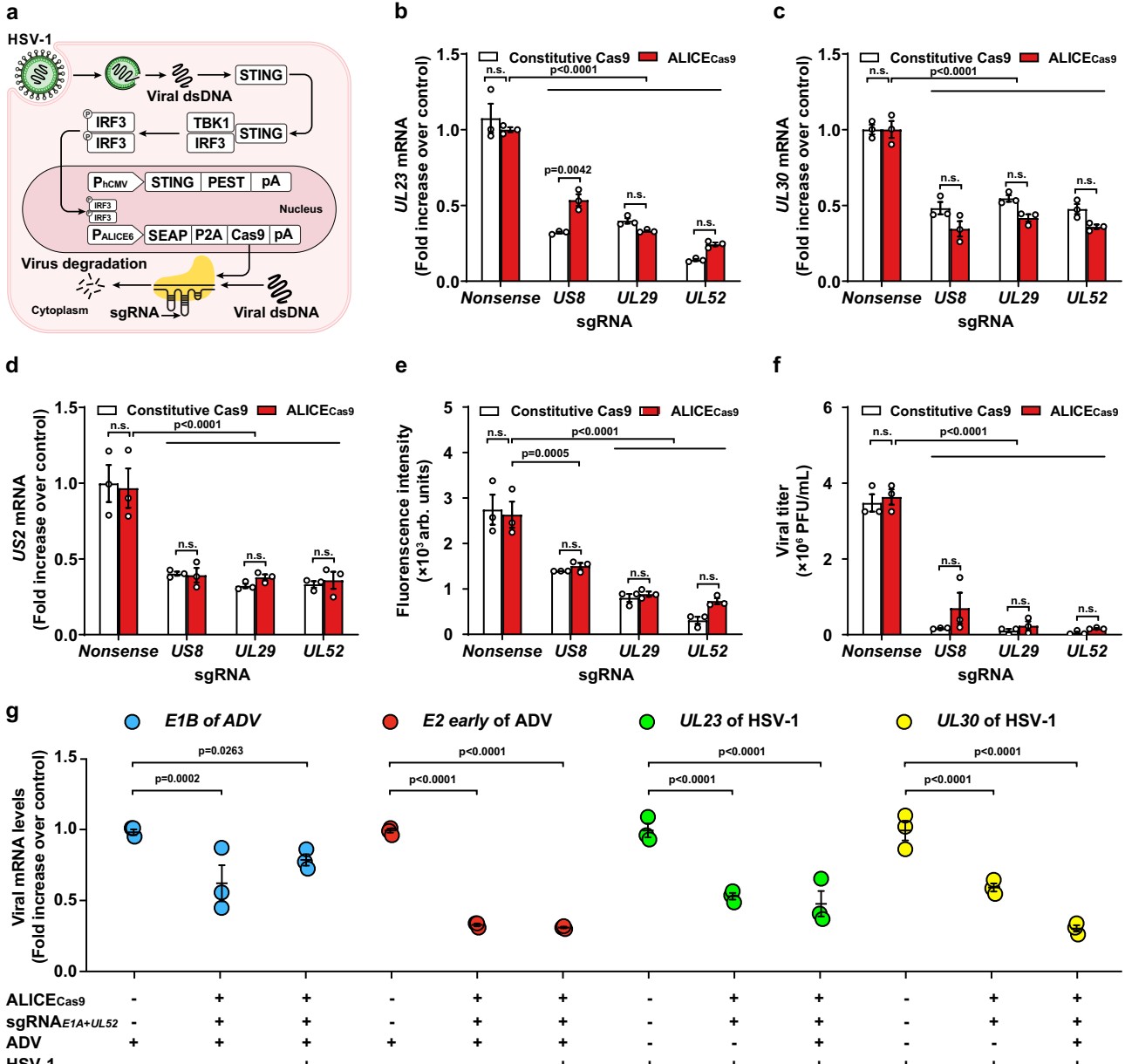

**Fig. 2 | Design and validation of an autonomous, intelligent, antiviral, virus-inducible immune-like sense-and-deletion cells (ALICE_Cas9 cells). a** Schematic illustration of the design principle of ALICE_Cas9 cells for autonomous sense-and-deletion of HSV-1. HSV-1 activates the virus sensor and initiates the expression of the Cas9 protein under the control of the virus-inducible promotor (P_ALICE6). HSV-1-inducible Cas9 protein inhibits viral replication by targeting and deleting highly conserved virus replication-related genes *US8/UL29/UL52* under the guidance of its corresponding sgRNA. **b** qPCR analysis of mRNA levels of the viral transcript *UL23* from EGFP-labeled HSV-1. Transfection of HSV-1-targeting sgRNA [highly conserved virus replication related gene: *US8* locus (pYW102); *UL29* locus (pYW172); *UL52* locus (pYW188), and a nonsense control locus (pWS68)] with pYW274/pYW169 or constitutive Cas9 (pYW54) was performed 20 h prior to EGFP-labeled HSV-1 infection (MOI = 5) in HEK-293T cells. The relative mRNA expression of *UL23, UL30* (**c**), and *US2* (**d**) was quantified by qPCR. All data were respectively normalized to the

*UL23, UL30, US2* gene expression levels in control group where HEK-293T cells co-transfected with pYW54/pWS68 were infected with EGFP-labeled HSV-1. White bars (Constitutive Cas9), red bars (ALICE_Cas9) in (**b**–**f**). **e** Fluorescence intensity of ALICE_Cas9. **f** EGFP-labeled HSV-1 titer in ALICE_Cas9. **g** qPCR analysis of *E1B/E2 early* genes from ADV, and *UL23/UL30* genes from HSV-1 in ALICE_Cas9 cells. ALICE_Cas9 cells transfected with sgRNAs targeting *E1A* of ADV and *UL52* of HSV-1 were incubated with a single virus (ADV or HSV-1) or both viruses (ADV and HSV-1). Blue circles (*E1B* of ADV), red circles (*E2 early* of ADV), green circles (*UL23* of HSV-1), yellow circles (*UL30* of HSV-1). The data in **b**–**g** represent the means ± SD; *n* = 3 independent experiments; *P* values in **b**–**g** were calculated by two-way ANOVA with Bonferroni's post hoc test; *P* values for all other groups versus *Nonsense* group in the same ALICE_Cas9 group, and between Constitutive Cas9 group and ALICE_Cas9 group in the same sgRNA; n.s. not significant. Source data are provided as a Source Data file.

(EGFP) reporter construct (Fig. 1c). ALICE_sen was functional in time and dosage-dependent manner (Fig. 1d, e). Moreover, we demonstrate that the ALICE_sen system was activated in five commonly used mammalian cell lines (Fig. 1f). Reversible induction kinetics of ALICE_sen was attained with the use of the antiviral drug acyclovir (ACV) (Supplementary Fig. 1e).

Type I interferons (IFN-α/β), involved in activation of innate immune signaling pathways, have broad-spectrum antiviral activity[38]. We engineered virus-inducible immune-like sense-and-clearance (ALICE_im) cells, mimicking the human innate immune system, where downstream genes driven by the synthetic promoter (P_ALICE6) were replaced with the human IFN-α or IFN-β. We next quantified

virus-induced IFN-α and IFN-β production, and antiviral efficacy. In the presence of four different viruses (HCV, HBV, ADV, and HSV-1), ALICE$_{im}$ cells can be induced to produce IFN-α (Fig. 1g–j) and IFN-β (Fig. 1k–n). The induced interferons can inhibit virus infection at the cellular level (Fig. 1o). These data demonstrate that ALICE$_{im}$ acts as an artificial innate immune system protecting the host organism through nonspecific immune defense and surveillance by the induction of interferons.

## Design and validation of autonomous, intelligent, virus-inducible immune-like cells with a Cas9 protein (ALICE$_{Cas9}$)

Previous studies have utilized platforms for virus detection and elimination based on the CRISPR gene editing system[39,40]. Notably, in these systems the Cas9 protein is constitutively expressed prior to virus infection. Constitutive expression of nuclease protein Cas9 has many side-effects, such as depletion of cellular resources, off-target effects, and increased risk of the emergence of CRISPR/Cas9-resistant escape mutant viruses[41]. To overcome these shortcomings, we engineered autonomous, intelligent, virus-inducible immune-like sense-and-deletion (ALICE$_{Cas9}$) cells. This rewired STING-TBK1-IRF3 pathway and P$_{ALICE6}$ promoter generates "immune-like cells" that can autonomously respond to HSV-1 and switch on the expression of Cas9 only upon the detection of viral dsDNA (Fig. 2a).

To generate ALICE$_{Cas9}$ cells, we replaced the SEAP reporter in ALICE$_{sen}$ with Cas9. The HSV-1-inducible expression of Cas9 was monitored by immunoblotting and a strong correlation was observed between Cas9 expression and HSV-1 multiplicity of infection (MOI) (Supplementary Fig. 2a). Our first iteration of ALICE$_{Cas9}$ immune-like cells was used for targeted deletion of the host cell gene C−C chemokine receptor type 5 gene (CCR5). In this iteration, we constitutively expressed an sgRNA targeting CCR5 (sgRNA$_{CCR5}$) (Supplementary Fig. 2b). In the presence of HSV-1, we achieved virus-induced, targeted deletion of the CCR5 locus of the human genome. We also successfully deleted an EYFP-fusion variant of an exogenous gene (d2EYFP) with sgRNA$_{d2EYFP}$ (Supplementary Fig. 2c).

After successfully demonstrating the HSV-1-inducible deletion of endogenous and exogenous genes using ALICE$_{Cas9}$ cells, we further demonstrated the self-sense-and-deletion of ALICE$_{Cas9}$ cells in the presence of HSV-1. Envelope glycoprotein E (US8), single-stranded DNA-binding protein (UL29), and helicase-primase primase subunit (UL52) have been demonstrated as functionally essential for viral propagation[25], therefore sgRNAs targeting these genes were selected to decrease viral replication.

To assess the HSV-1 life-cycle[42,43], thymidine kinase (UL23), DNA polymerase catalytic subunit (UL30), and virion protein US2 (US2) genes representing immediate early, early and late stages of HSV-1 replication, respectively, were monitored (Fig. 2b–d). ALICE$_{Cas9}$ cells demonstrated antiviral activity in the presence of EGFP-labeled HSV-1 that encodes EGFP reporter gene, as measured by HSV-1 viral RNA levels (Fig. 2b–d), virus infected cells (Fig. 2e), and live virus particles (Fig. 2f and Supplementary Fig. 2d). ALICE$_{Cas9}$ cells showed comparable antiviral effects, compared with the constitutive Cas9 system (Fig. 2b–f and Supplementary Fig. 2d). Since CRISPR relies on base-pairing of sgRNAs to the target nucleotide sequences[44], multiple sgRNAs targeting different viruses can be incorporated in ALICE$_{Cas9}$. ALICE$_{Cas9}$ cells transfected with tandem sgRNAs targeting both E1A gene of ADV and UL52 gene of HSV-1, demonstrated antiviral activity (ADV and HSV-1) as measured by ADV and HSV-1 RNA levels (Fig. 2g). These data indicating that the ALICE$_{Cas9}$ system can be designed to target and destroy multiple viruses by expressing tandem sgRNAs.

## Design and validation of autonomous, intelligent, virus-inducible immune-like cells with a neutralizing antibody (ALICE$_{Ab}$)

Next, we assessed an ALICE antibody system. We used a known HSV-1 human monoclonal neutralizing antibody E317 (mAb E317Ab)[45] in HSV-1-inducible immune-like designer (ALICE$_{Ab}$) cells. Specifically, ALICE$_{Ab}$ cells express P$_{hCMV}$-driven STING-PEST (pYW274) and P$_{ALICE6}$-driven His-tag E317Ab (pYW364) (Supplementary Fig. 3a). After confirming the functional role of E317Ab in HSV-1 neutralization, (Supplementary Fig. 3b), we explored the antiviral kinetics of ALICE$_{Ab}$ cells against HSV-1 (Supplementary Fig. 3c). ALICE$_{Ab}$ cells decreased viral load after infection with EGFP-labeled HSV-1 at an MOI of 1-5 (Supplementary Fig. 3c).

As reported by previous study[46], a cocktail of antibodies REGN10989 and REGN10987 can neutralize SARS-CoV-2[47]. We generated a SARS-CoV-2-responsive ALICE$_{Ab}$ system, containing a P$_{hCMV}$-driven STING protein (pYW274) and a plasmid to drive the antibody production (pYW406, P$_{ALICE6}$-driven REGN10989, and REGN10987) (Supplementary Fig. 4a–c). We found that ALICE$_{Ab}$ was able to reduce SARS-CoV-2 infection by ~70.3 ± 4.3% (Supplementary Fig. 4c).

## Design and validation of autonomous, intelligent, virus-inducible immune-like sense-and-destroy cells (ALICE$_{Cas9+Ab}$)

Following proof-of-concept studies with antiviral ALICE$_{Cas9}$ and ALICE$_{Ab}$ cells, we extended our work by establishing a more robust self-sensing and inhibition system. We constructed ALICE$_{Cas9+Ab}$ cells harboring Cas9 and E317Ab for synergistic activity (Fig. 3a). After generating and confirming the function of ALICE$_{Cas9}$-transgenic stable cell lines (HEK$_{ALICE-SEAP-Cas9}$) (Supplementary Fig. 5) and ALICE$_{Cas9+Ab}$-transgenic stable cell lines (HEK$_{ALICE-Cas9-E317Ab}$) (Supplementary Fig. 6), quantitative profiling and immunoblotting confirmed that the presence of HSV-1 strongly induced the expression of Cas9 and mAb E317Ab in the HEK$_{ALICE-Cas9-E317Ab}$ cells.

We compared the antiviral performance of single-output immune-like designer cells (ALICE$_{Cas9}$ and ALICE$_{Ab}$ cells) against the dual-output immune-like designer ALICE$_{Cas9+Ab}$ cells. ALICE$_{Cas9+Ab}$ system outperformed ALICE$_{Cas9}$ and ALICE$_{Ab}$ system, exhibiting potent and synergistic inhibition of viral replication in HEK-293T cells (Fig. 3b). Subsequent quantitation of SEAP production demonstrated that the modules of ALICE system do not have negative effects either on overall gene expression capacity of the transfected cells (Supplementary Fig. 7). Assessment of the long-term antiviral effects of ALICE$_{Cas9+Ab}$ cells revealed continuous expression of both the Cas9 and E317Ab proteins in the presence HSV-1 over the course of one week (Fig. 3c). We also tested the performance of ALICE$_{Cas9+Ab}$ cells against a known HSV-1 antiviral drug ACV. We found that ALICE$_{Cas9+Ab}$ cells exerted similarly potent antiviral effects against viral replication in HEK-293T cells as a high-dose ACV (50 μM) and found that ALICE$_{Cas9+Ab}$ cells significantly outperformed low-dose ACV (10 μM). The antiviral outputs from ALICE$_{Cas9+Ab}$ cells are specifically induced by the presence of the virus and comprise two orthogonal modes of action against HSV-1 (Fig. 3c). Development of drug resistance by viruses is a major cause for concern[48] and our findings clearly highlight the attractive potential of multi-output therapeutics based on pathogen-responsive cells.

Furthermore, we performed two types of Transwell®-based assays to characterize antiviral performance of ALICE$_{Cas9+Ab}$ cells: protection against viral spread among host cells (HEK-293T cells) and protection against infection of the designer cells (ALICE$_{Cas9+Ab}$) themselves. The presence of functioning ALICE$_{Cas9+Ab}$ cells strongly inhibited the spread of HSV-1 among HEK-293T cells (Fig. 3d, e). Infected HEK-293T cells were added to functioning ALICE$_{Cas9+Ab}$ or ALICE-like cells and the signal for EGFP-labeled HSV-1 was dramatically lower in the functional ALICE$_{Cas9+Ab}$ cells at 48 h post-infection (hpi) (Fig. 3f, g), confirming the self-protection capacity of the ALICE$_{Cas9+Ab}$ cells.

## Sense-and-destroy against HSV-1 mediated by ALICE in mice

We performed a pilot study to rigorously assess the feasibility of ALICE systems in vivo, by establishing a series of HSV-1-infected mouse models. To examine the potential of ALICE systems to prevent viral infections, we delivered immune-like cells containing our single- and

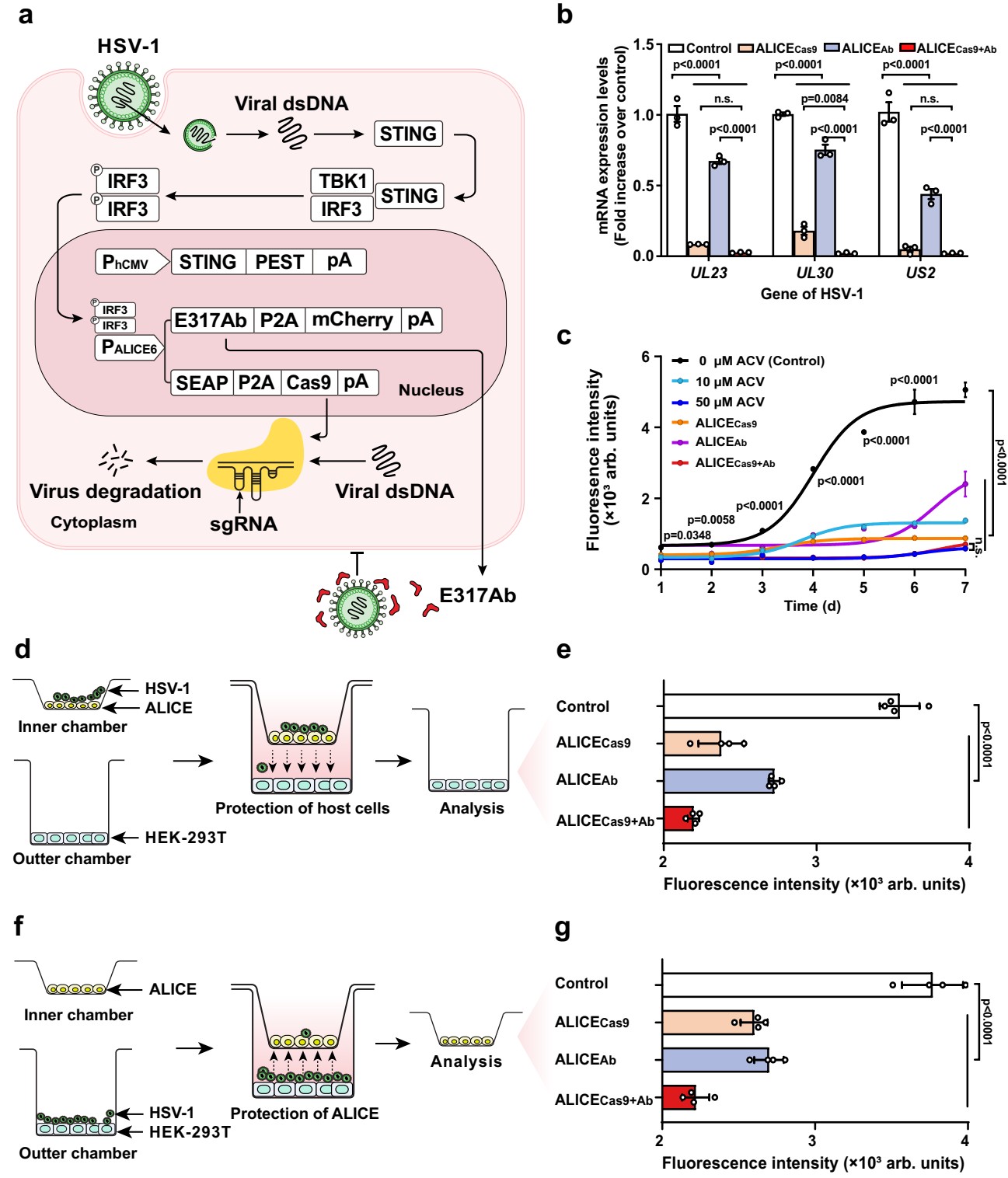

dual-output ALICE systems into mice and evaluated the antiviral effects (Fig. 4a). From a biocompatibility perspective, we selected hyaluronic acid-based hydrogels, as graft materials for immune-like cells, that are made of hydrophilic polymer chains, cross-linked to swell and retain their three-dimensional structure without dissolving[49]. We embedded control HEK-293T cells, both types of the single-output ALICE$_{Ab}$/ALICE$_{Cas9}$ cells, and the dual-output ALICE$_{Cas9+Ab}$ cells, into hydrogel scaffolds and introduced the matrix into the abdomen of mice via intraperitoneal transplantation. Subsequently, we challenged mice with HSV-1 ($2 \times 10^7$ PFU, plaque-forming unit) at 20 h post-transplantation.

The mice were sacrificed at day 2, 4, and 6 post-infection (dpi) to monitor viral titers in organs including liver, spleen, and kidney (Fig. 4a). We determined that ALICE$_{Cas9+Ab}$ had the best antiviral effect for up to 6 days, then further examined the antiviral activity in this system. E317Ab expression in ALICE$_{Cas9+Ab}$ cells-implanted mice was significantly higher when challenged with virus (Fig. 4b). Likewise, we excised the hydrogels and confirmed that Cas9 expression in mice bearing scaffolds with ALICE$_{Cas9+Ab}$ cells was much higher when challenged with HSV-1, compared with the non-challenge group (Fig. 4c). As shown in Supplementary Fig. 8, no significant difference was found

**Fig. 3 | Viral sense-and-destroy function in ALICE system. a** Schematic illustration of the design principle of ALICE$_{Cas9+Ab}$ system to autonomously sense-and-destroy of HSV-1. HSV-1 activates the ALICE$_{Cas9+Ab}$ system and ultimately initiates the expression of Cas9 and E317Ab under the control of the virus-inducible promotor (P$_{ALICE6}$). HSV-1-inducible Cas9 protein inhibits viral replication by deleting highly conserved sites on virus replication-related genes, while the expression of a recombinant monoclonal antibody that targets an epitope on glycoprotein D of HSV-1 (E317Ab) blocks viral infection. **b** qPCR analysis of EGFP-labeled HSV-1 mRNA levels in the ALICE system. White bars (Control), brown bars (ALICE$_{Cas9}$), blue-gray bars (ALICE$_{Ab}$), red bars (ALICE$_{Cas9+Ab}$) in (**b, e, g**). *P* values for all other groups versus Control group in the same gene. **c** The comparison study of antiviral effects between acyclovir and ALICE system in mammalian cells. *P* values for ALICE$_{Cas9+Ab}$ group versus Control group, and 50 μM ACV group on the same day. **d** Schematic for the protection function of ALICE system to host cells. Immune-like designer cells containing ALICE system were infected with EGFP-labeled HSV-1 (MOI = 1) for 3 h and then seeded on the inner chamber membrane of a Transwell® apparatus. Before this, HEK-293T cells had been seeded on the bottom of the outer chamber. EGFP expression of HEK-293T cells seeded on the bottom was profiled (**e**). **f** Schematic for the protection function of ALICE system to itself. HEK-293T cells were seeded on the bottom of the outer chamber and infected with EGFP-labeled HSV-1 (MOI = 1) for 3 h. Immune-like designer cells containing ALICE system were then seeded on the inner chamber membrane and embedded into the outer chamber. EGFP expression of immune-like designer cells containing ALICE system seeded on the inner chamber were profiled (**g**). The data in **b, c, e**, and **g** are expressed as means ± SD; *n* = 3 or 4 independent experiments in **b, e**, and **g**; *n* = 3 independent experiments in (**c**); *P* values were calculated by two-way ANOVA with Bonferroni's post hoc test; n.s. not significant. Source data are provided as a Source Data file.

among three groups, including non-treated mice (WT), unmodified-hydrogel-implanted mice (Control), and designer-hydrogel-implanted mice (ALICE$_{Cas9+Ab}$), indicating that hydrogel-based transplantation alone did not invoke host-mediated, foreign-body responses.

The organs of mice that received the control HEK-293T cells had high levels of viral RNAs including *UL23/US2* at the various dpi. In contrast, lower level of viral RNAs was detected in organs of the animals given the dual-output ALICE system. The viral RNA levels in organs were at intermediate levels upon transplantation with the single-output ALICE systems, with the mAb system slightly outperforming the Cas9 system (Fig. 4d–l). The superior performance of the dual-output ALICE system was increasingly evident over time (Fig. 4d–l), and the most pronounced effect was detected in the liver at 6 days post-transplantation (Fig. 4j–l).

### Long-term sense-and-destroy against HSV-1 mediated by ALICE$_{Cas9+sgRNAs+Ab}$ in mice

We subsequently evaluated the long-term antiviral functionality of ALICE$_{Cas9+Ab}$ cells in mice. The hydrogel implantation of the stable dual-output HEK$_{ALICE-Cas9-sgRNAs-E317Ab}$ cells (ALICE$_{Cas9+sgRNAs+Ab}$ cells) into mice was the same as described in Fig. 4a (Fig. 5a). To determine the longevity of ALICE$_{Cas9+sgRNAs+Ab}$ cells activity in vivo, we challenged the hydrogel-implanted mice with HSV-1 (2 × 10$^7$ PFU) at 28 days post-transplantation. We sacrificed mice at 30 days post-transplantation to assess viral titers in organs including liver, spleen, and kidney (Fig. 5a). The location and shape of hydrogels was recorded before and after transplantation (Supplementary Fig. 9). Our observations were consistent with previously published studies where the viability of cells in hydrogels are maintained for at least 30 days[50]. Cas9 expression in ALICE$_{Cas9+sgRNAs+Ab}$ cells-implanted mice was much higher when challenged with HSV-1, compared with the non-challenged group (Fig. 5b) and similar findings were observed for the E317Ab expression (Fig. 5c).

We then monitored HSV-1 mRNA levels at 30 days post-transplantation in mouse organs positioned near the transplant sites (liver, spleen, and kidney) (Fig. 5d, e). Our results showed that the organs derived from unmodified-HEK-293T cells-implanted mice had higher levels of viral RNAs and titers, compared with ALICE$_{Cas9+sgRNAs+Ab}$ cells-implanted mice (Fig. 5d–f). To determine shuttle efficacy of HSV-1 in hydrogels, we used quantitative PCR with reverse transcription (RT-qPCR) to quantify copies of HSV-1 mRNA in unmodified-HEK-293T-hydrogels challenged with HSV-1 (Control), normalized to hydrogels without HSV-1. The average level of HSV-1 mRNA (*UL23* and *US2*) in unmodified-HEK-293T-hydrogels challenged with HSV-1 was 1000–3000 folds higher than hydrogels without HSV-1 (Fig. 5g). There is consensus that severe virus infection results in excessive virus-induced inflammation mediated by the infiltration of inflammatory cells, including T cells (both CD4$^+$ and CD8$^+$), polymorphonuclear leukocytes and macrophages[5]. Indeed, HSV-1 infection provokes the expression of the inflammatory molecules IL-6, CCL5, CXCL10, TNF-α, and IFN-α, which was blocked after

ALICE$_{Cas9+sgRNAs+Ab}$ cells-hydrogel transplantation (Fig. 5h). Moreover, the innate immune response of mice is important for antiviral effects. Our results showed that there is no significant difference in the expression of Immunoglobulin G (IgG) between unmodified-HEK-293T-hydrogels implanted mice (Control) and ALICE$_{Cas9+sgRNAs+Ab}$ cells-hydrogels-implanted mice (treated group), when mice were challenged with HSV-1 (Fig. 5i).

### Inhibition of HSV-1 transmission mediated by ALICE$_{Cas9+Ab}$ in mice

During HSV-1 latency, the viral genome is harbored in peripheral neurons in the absence of infectious virus but has the potential to restart infection[51]. To mimic an organ-transplant recipient who received a latent HSV-1-infected organ, we developed a mouse model where mice received HSV-1-infected ALICE$_{Cas9+Ab}$ cells. To verify the inhibition of viral spread efficacy, we evaluated the inhibition of viral transmission mediated by ALICE$_{Cas9+Ab}$, which were infected with HSV-1 before implantation into mice. We used the engineered stable cell line HEK$_{ALICE-Cas9-E317Ab}$ infected with HSV-1 and stabilized the cells in hyaluronic acid-based hydrogel scaffolds. The hydrogel scaffolds were intraperitoneally transplanted into mice as a central point of infection for surrounding organs, thereby establishing infection in the mice (Supplementary Fig. 10a). We first examined the inducible expression of ALICE proteins by excising the hydrogels containing uninfected HEK$_{ALICE-Cas9-E317Ab}$ cells or HSV-1-infected HEK$_{ALICE-Cas9-E317Ab}$ cells at 6 days post-transplantation. Total proteins were extracted from these scaffolds and immunoblotting revealed abundant Cas9 expression in the scaffolds bearing the HSV-1-infected HEK$_{ALICE-Cas9-E317Ab}$ cells but very low background in the control scaffolds. These results highlighted both the virus inducibility of ALICE and a low level of leaky Cas9 expression in the absence of the virus (Supplementary Fig. 10b). Similar observations were obtained for E317Ab expression (Supplementary Fig. 10c). Further, analyses of serum collected at 2-, 4-, and 6-days post-transplantation revealed that the IL-6 levels were consistently higher in the animals transplanted with the HSV-1-infected HEK-293T cells than the animals with the HSV-1-infected HEK$_{ALICE-Cas9-E317Ab}$ cells (Supplementary Fig. 10d), indicating the antiviral effects of ALICE$_{Cas9+Ab}$.

We then monitored the presence of the viral RNA (*UL23* and *US2*) level in mouse organs positioned near the transplant site (liver, spleen, and kidney) (Supplementary Fig. 10e, f, h, i, k, l). The organs of animals transplanted with infected HEK-293T hydrogel scaffolds exhibited significantly higher viral RNA levels and titers than HEK$_{ALICE-Cas9-E317Ab}$ hydrogel scaffolds (Supplementary Fig. 10e–m).

### Sense-and-destroy against HSV-1 mediated by ALICE$_{Cas9+Ab}$ in a virus-infected mouse model

To examine the antiviral potential of immune-like designer cells ALICE$_{Cas9+Ab}$ in vivo, we used a virus-infected mouse model where mice were first infected with HSV-1 (2 × 10$^7$ PFU) via intraperitoneal injection. At 20 hpi, the engineered stable HEK$_{ALICE-Cas9-E317Ab}$ cells were

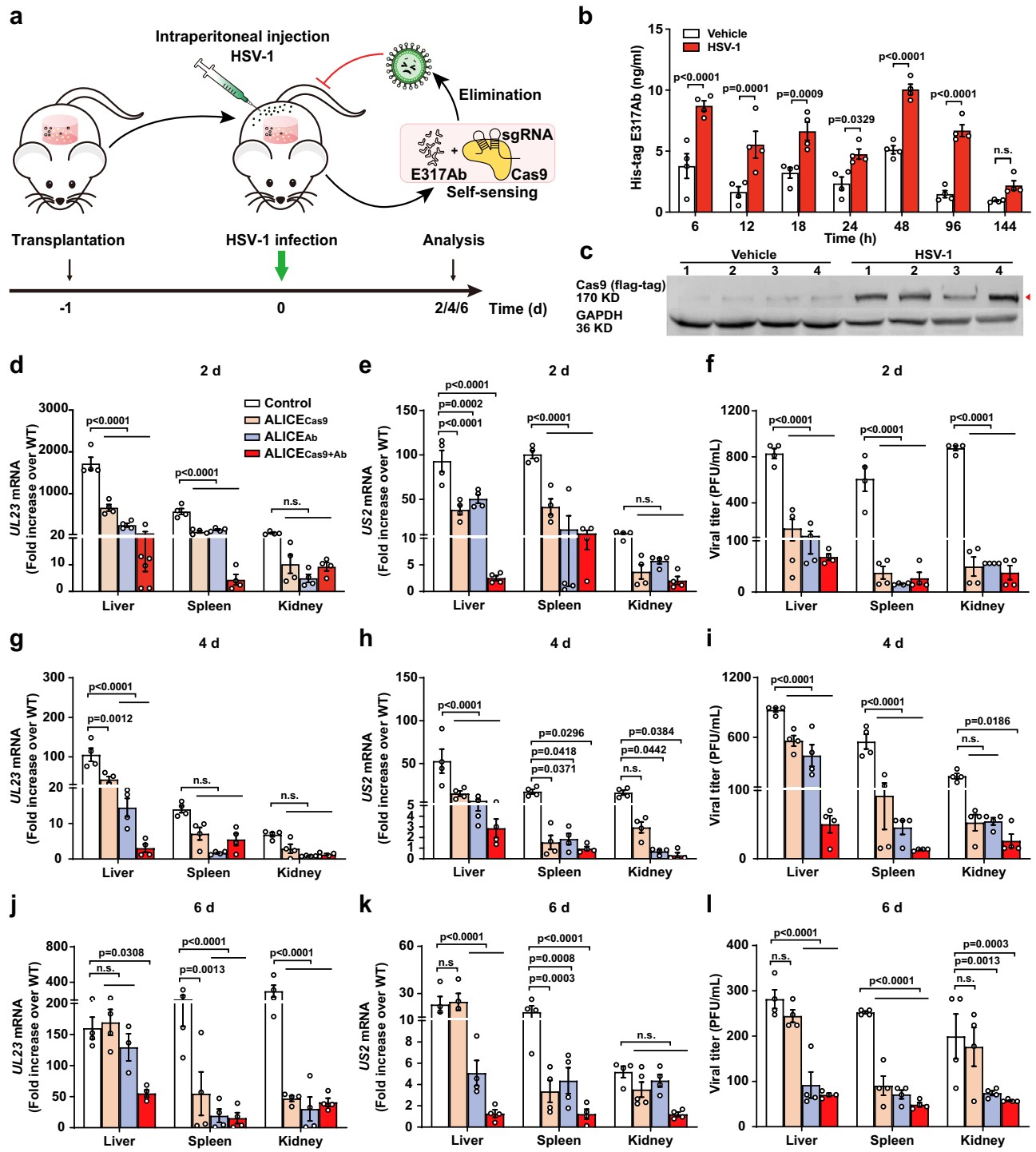

encapsulated into hyaluronic acid-based hydrogel scaffolds, which were intraperitoneally transplanted into mice as a self-sensing antiviral device, thereby decreasing infection in the mice (Fig. 6a). We examined the inducible protein expression from ALICE$_{Cas9+Ab}$ by excising the ALICE$_{Cas9+Ab}$ hydrogels from HSV-1-uninfected or -infected mice at 6 days post-transplantation. Total proteins were extracted from these scaffolds: immunoblotting revealed abundant Cas9 expression in HSV-1-infected mice containing ALICE$_{Cas9+Ab}$ scaffolds, but very low levels in the HSV-1-uninfected mice (Fig. 6b). These results indicated that virus present in the abdominal cavity can freely shuttle into the scaffolds, where it can initiate an antiviral response from the ALICE$_{Cas9+Ab}$ hydrogels and triggered the E317Ab expression in mice containing ALICE$_{Cas9+Ab}$-hydrogel implant (Fig. 6c).

We then determined the levels of HSV-1 viral RNA (*UL23* and *US2*) at 6 days post-transplantation in mouse organs positioned near the transplant sites (liver, spleen, and kidney) (Fig. 6d, e). Our results showed that the organs of unmodified-HEK-293T cells-implanted mice exhibited strong expression of viral RNAs and viral titers, compared with ALICE$_{Cas9+Ab}$ cells-implanted mice, notably where all mice were challenged with HSV-1 ($2 \times 10^7$ PFU) (Fig. 6d–f).

**Long-term sense-and-destroy against HSV-1 mediated by AAV-ALICE$_{SaCas9+Ab}$ in a herpetic simplex keratitis mouse model**
After assessing the antiviral efficacy of the ALICE system in mouse models via cell therapy, we next investigated the antiviral potential of ALICE$_{Cas9+Ab}$ delivered by AAV-vector in a HSK mouse model, which

**Fig. 4 | Sense-and-destroy against HSV-1 mediated by ALICE in mice. a** Schematic illustration of single- and dual-output ALICE systems for autonomous sense-and-destroy against HSV-1 in mice. Four types of cells, including (1) three sgRNAs (pYW102/pYW172/pYW188, respectively targeting *US8/UL29/UL52*)-transgenic HEK$_{ALICE-SEAP-Cas9}$ (ALICE$_{Cas9}$), (2) pcDNA3.1-transgenic HEK$_{ALICE-Cas9-E317Ab}$ (ALICE$_{Ab}$), (3) pYW102/pYW172/pYW188-transgenic HEK$_{ALICE-Cas9-E317Ab}$ (ALICE$_{Cas9+Ab}$), or (4) pcDNA3.1-transgenic HEK-293T cells (Control), were encapsulated into hydrogel-based scaffolds and transplanted into the abdomen of mice via intraperitoneal surgery. At 20 h post-transplantation of the hydrogel implants, HSV-1 ($2 \times 10^7$ PFU) was intraperitoneally injected into each mouse. The antiviral effects of the single- and dual-output ALICE systems in mice were evaluated by detecting residual viral titers in indicated organs (liver, spleen, kidney) at 2, 4 and 6-days post HSV-1 injection (**d–l**). **b** Assay of HSV-1-inducible E317Ab expression and **c** western blot analysis of HSV-1-inducible Cas9 expression in hydrogel-scaffolds. Mice were transplanted with hydrogel-scaffolds containing pYW102/pYW172/pYW188-transgenic HEK$_{ALICE-Cas9-E317Ab}$ cells, infected with HSV-1 ($2 \times 10^7$ PFU, HSV-1 group) or uninfected (0, Vehicle group). The red arrowhead indicates the expected Cas9 band. **d** qPCR assay of HSV-1 *UL23/US2* mRNA in mice. HSV-1 mRNA levels were performed in isolated liver/spleen/kidney using specific *UL23/US2* primers listed in Supplementary Table 2 at 2 (**d, e**), 4 (**g, h**), and 6 (**j, k**) days post HSV-1 injection. White bars (Control), brown bars (ALICE$_{Cas9}$), blue-gray bars (ALICE$_{Ab}$), red bars (ALICE$_{Cas9+Ab}$) in (**d–l**). **f** Viral titers in mice. The mice were processed as described in (**a**). Virus in isolated tissues, such as liver, spleen, kidney, were titrated at 2 (**f**), 4 (**i**), and 6 (**l**) days post HSV-1 injection. The relative expression was calculated using the ΔΔct method based on the expression levels of viral genes *UL23/US2* in organs of wild-type mice (WT). *P* values for all other group versus Control group in the same tissue. Data in **d, e, g, h, j, k** are normalized to wild-type mice (WT); Numbers 1–4 represents four indenpent mice in (**c**); Data in **b, d–l** are expressed as means ± SEM; *n* = 4 mice; *P* values were calculated by two-way ANOVA with Bonferroni's post hoc test; n.s. not significant. Source data are provided as a Source Data file.

mimics natural HSV-1 infection. As reported by previous studies, AAV-mediated delivery of meganucleases[52], *Streptococcus pyogenes* Cas9 (SpCas9), or *Staphylococcus aureus* Cas9 (SaCas9)[53] mediated highly efficient gene editing of HSV-1 from TG. AAV has shown great promise for gene delivery in vivo as well as in TG neurons. Due to the packaging limit of AAV (<4500 bp), SaCas9 (3156 bp) is more suitable for AAV-mediated delivery than SpCas9 (4101 bp), which leaves no space for AAV-ALICE$_{SaCas9+Ab}$ regulatory elements. As an expression of SaCas9 with HSV-1 immediate-early regulatory protein ICP4-targeted sgRNA can lead to the decrease of the fluorescence intensity of EGFP trigged by HSV-1 replication, we confirmed that expression of the CRISPR-SaCas9 system could reduce HSV-1 infection (Supplementary Fig. 11a, b).

To test whether the AAV-ALICE$_{SaCas9+Ab}$ system can eliminate HSV-1 in TG, we established an HSK mouse model. The AAV-ALICE$_{SaCas9+Ab}$ system consists of two AAV elements: firstly, the AAVrh10-ALICE$_{SaCas9}$ carries an HSV-1-induced SaCas9 and a constitutive P$_{U6}$-driven expression of HSV-1-targeted *ICP4* sgRNA ($5 \times 10^{11}$ PFU/mouse); secondly, the AAV1-ALICE$_{Ab}$ carries an HSV-1-induced E317Ab-P2A-nanoLuc and a constitutive P$_{hCMV}$-driven expression of STING-PEST ($5 \times 10^{11}$ PFU/mouse). Wild-type mice were injected with AAV-ALICE$_{SaCas9+Ab}$ system via retro-orbital (RO) injection. At 6 days post AAV-ALICE$_{SaCas9+Ab}$ system injection, challenge with HSV-1 infection ($9 \times 10^5$ PFU per eye) was conducted following corneal scarification at 0 and 20 days to mimic the HSK mice (Fig. 7a). Body weight was recorded daily. Significant weight loss was observed for the HSV-1-treated mice without the AAV-ALICE$_{SaCas9+Ab}$ system, and a clear difference was observed between the HSV-1 infected mice in the presence or absence of the AAV-ALICE$_{SaCas9+Ab}$ system (Fig. 7b). Next, we evaluated the virus titers in isolated corneas, TG and brain, at 14- and 25-days post initial HSV-1 infection, and found significantly lower titers in all tested organs isolated from AAV-ALICE$_{SaCas9+Ab}$ system-treated mice than PBS-treated mice (Fig. 7c–h).

Finally, we examined whether RO injection of AAV-ALICE$_{SaCas9+Ab}$ system induces HSV-1-responsive nanoLuc (Fig. 7i) and E317Ab (Fig. 7j) in the bloodstream. We observed significantly higher HSV-1-responsive nanoLuc and E317Ab in the AAV-ALICE$_{SaCas9+Ab}$ system-treated mice challenged with HSV-1 ($9 \times 10^5$ PFU per eye) than those without HSV-1. Moreover, HSV-1 infection induced expression of the inflammatory molecules IL-6, CCL5, CXCL10, TNF-α, and IFN-α, which was blocked after AAV-ALICE$_{SaCas9+Ab}$ system treatment at 5-, 14-, and 25-days post first HSV-1 infection (Fig. 7k–o). Notably, our results showed that there is no significant difference in the expression of IgG between PBS-treated mice (Control) and AAV-ALICE$_{SaCas9+Ab}$ system-treated mice (Treated group), where all mice were challenged with HSV-1 (*n* = 4 mice; non-significant, two-tailed Student's *t* tests) (Supplementary Fig. 11c). Taken together, these results suggest that the administration of AAV-ALICE$_{SaCas9+Ab}$ system significantly reduced the manifestation of disease severity in the HSK mouse model.

## Discussion

Here, we showed that the immune-like designer cells (ALICE cells) harboring a STING-mediated genetic circuit can effectively detect multiple viruses and we demonstrated the utility of ALICE systems for detection and inhibition/elimination of viruses based on nuclease-mediated cleavage and antibody-mediated virus neutralization. The ALICE system can detect and mitigate infections before symptoms appear, which is particularly beneficial for immune compromised patients for whom infections can be life-threatening[54].

The availability of a closed-loop gene network to regulate therapeutic agent release in mammalian cells could facilitate implementation of novel anti-infection therapies. Protein therapeutics like our examples of antiviral cytokines (human IFN-α or IFN-β)[38], Cas9[55], and the neutralizing antibody E317Ab[56] represent promising alternatives to conventional antiviral drugs[57,58]. As engineered cells can accommodate many modifications to their genomes, entire biosynthetic pathways for the release of potent antiviral molecules could be engineered into these immune-like designer cells. Therefore, the ALICE system can deploy alternative therapies that are collectively less likely to induce the development of antiviral resistance. Beyond demonstrating a clinically relevant level of anti-HSV-1 potency, our results illustrate a potential to overcome some of the known problems with the emergence of ACV-resistant virus strains (Fig. 3c)[59].

The ALICE$_{Cas9}$ system could be modified to achieve high viral cleavage efficiency based on alternative sgRNAs that target other viral regions or even target distinct sites of multiple viruses. In addition, the virus-specific neutralizing antibody output modules of ALICE$_{Ab}$ can be changed to express the best-available antibodies to increase potency and/or therapeutic half-life or target various viruses, such as SARS-CoV-2 (Supplementary Fig. 4). These modifications can be combined for new outputs from ALICE$_{Cas9+Ab}$ for combination therapies to achieve synergistic antiviral effects.

There is no cure available for HSV-1 and once a person is infected, the virus remains dormant in the TG, with the potential to reactivate. The leading treatment for HSV-1 infection is acyclovir, a purine nucleoside analog that can inhibit viral replication. These antiviral treatment options are given in combination with topical steroids, which may not be suitable for long-term use due to their side effects, such as secondary glaucoma, infection, and cataracts. This is a major limitation when it comes to treating HSK, which is often recurrent and requires extended and as well prophylactic treatment regimens[60]. Gene therapy became possible through the advances of genetics and bioengineering that enabled manipulating vectors for delivery of extrachromosomal material to target cells[61]. Therefore, we use the engineered AAV vectors to deliver all DNA elements of our ALICE$_{SaCas9+Ab}$ into targeted tissues for a long-term, cocktail HSK therapy.

The success of ALICE cells to control HSV-1 infection in mice, delivered at different stages of virus infection either before, during, or

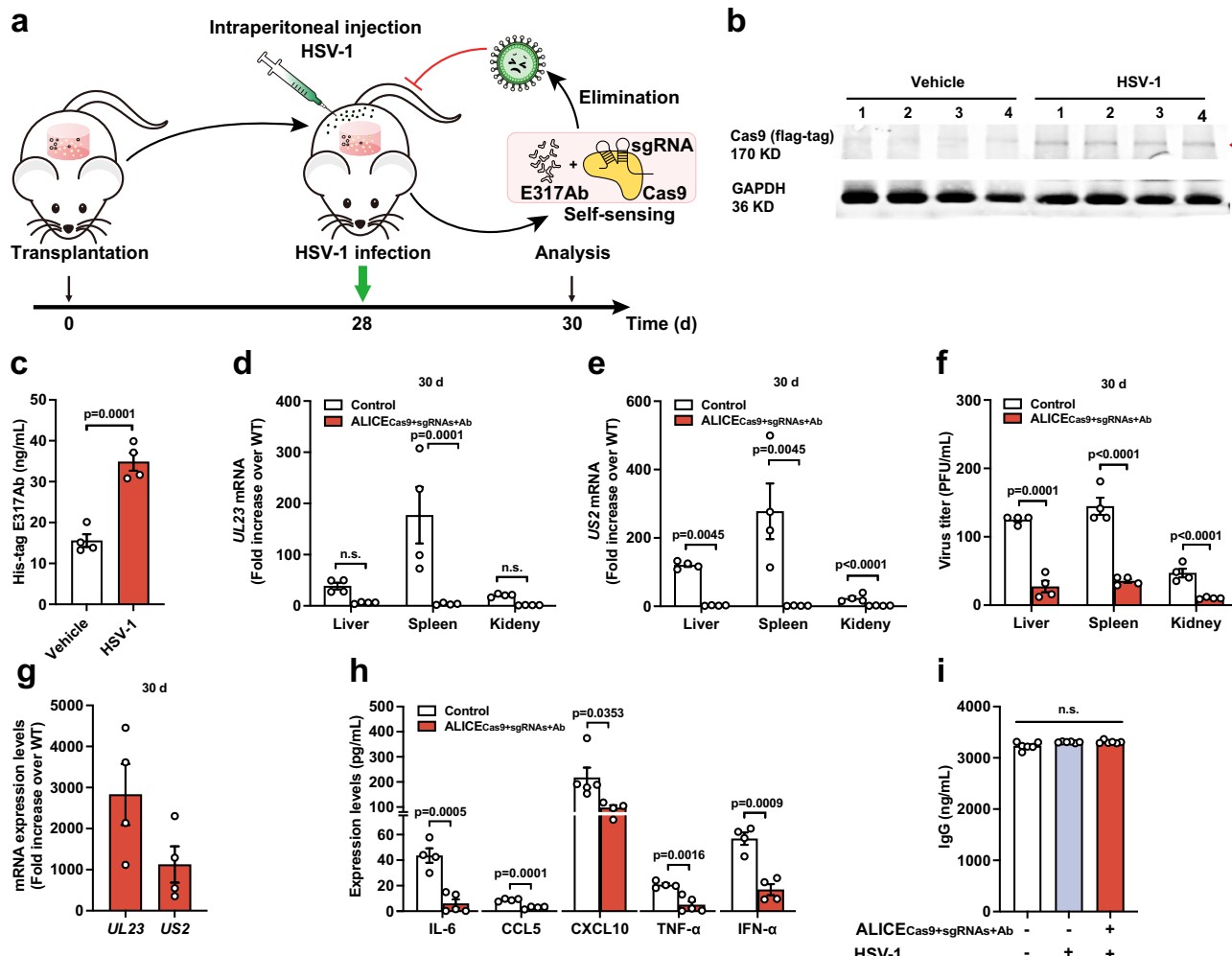

**Fig. 5 | Long-term sense-and-destroy against HSV-1 mediated by ALICE_{Cas9+sgRNAs+Ab} in mice. a** Schematic illustration of ALICE_{Cas9+sgRNAs+Ab} system for autonomous sense-and-destroy against HSV-1 in mice. HEK-293T cells stably integrated with ALICE_{Cas9+sgRNAs+Ab} (HEK_{ALICE-Cas9-sgRNAs-E317Ab} cells) or pcDNA3.1-transgenic HEK-293T cells (Control) were encapsulated into hydrogel-based scaffolds and transplanted into mice abdomens via intraperitoneal surgery. At 28 d post-transplantation, HSV-1 ($2 \times 10^7$ PFU) was intraperitoneally injected into each mouse. The antiviral effects of the ALICE_{Cas9+sgRNAs+Ab} in mice was evaluated by detecting residual viral titers in the indicated organs (liver, spleen, kidney) at 2 days post HSV-1 infection. **b** Western blot analysis of HSV-1-inducible Cas9 expression in hydrogel-based scaffolds. The red arrowhead indicates the expected Cas9 band. Number 1–4 represents four independent samples. **c** Validation of HSV-1-inducible E317Ab expression in mice. The E317Ab level in the bloodstream from ALICE_{Cas9+sgRNAs+Ab}-treated mice, infected with or without HSV-1, was quantified at 30 days post-transplantation. **d**, **e** qPCR analysis of HSV-1 *UL23/US2* mRNA levels in

mice. **f** Viral titers in mice. Virus in isolated tissues (liver, spleen, kidney) from ALICE_{Cas9+sgRNAs+Ab}-treated mice, infected with or without HSV-1, were titrated at 30 days post-transplantation. **g** qPCR analysis of HSV-1 *UL23/US2* mRNA in hydrogel implants at 30 days post-transplantation. **h** Cytokine levels in mice. Cytokines (IL-6, CCL5, CXCL10, TNF-α, and IFN-α) in the blood from HSV-1-infected mice implanted with or without ALICE_{Cas9+sgRNAs+Ab} were analyzed by flow cytometry at 30 days post-transplantation. **i** IgG expression in mice. Mice implanted with or without ALICE_{Cas9+sgRNAs+Ab} were infected with or without HSV-1. IgG levels in the bloods were analyzed using an IgG ELISA at 30 days post-transplantation. Data in **d**, **e**, and **g** are normalized to wild-type mice (WT); Data in **c**–**i** are expressed as means ± SEM; *P* values in **c**, **h** were calculated by two-tailed unpaired *t*-test; *P* values in **d**–**f** were calculated by two-way ANOVA with Bonferroni's post hoc test; *P* values in **i** were calculated by one-way ANOVA followed by a Dunnett's post hoc test; *n* = 4 mice in **c**–**g**, *n* = 4 or 5 mice in (**h**), *n* = 6 mice in (**i**). Source data are provided as a Source Data file.

after the transplantation of hydrogel scaffolds, via cell therapy or into HSK mice via gene therapy, warrants further investigation to explore ALICE's utility against other viruses[62].

It is important to note that the current study was designed to deliver a specific output in response to HSV-1 infection; the ALICE system could in theory, detect any dsDNA virus infection. Moreover, considering that many highly pathogenic viruses are RNA viruses (SARS-CoV-2, Nipah virus, Ebola virus, and influenza virus), application of the ALICE concept should be explored for the development of surveillance and treatment systems against RNA viruses. This could be achieved by replacing the STING-based sensor of our current ALICE system with an RNA sensor such as the retinoic-acid-inducible gene I (RIG-I). Successful activation of our current STING-based ALICE sensor

by SARS-CoV-2 and DENV-2 could have resulted from the release of nuclear or mitochondrial DNA due to cellular stress or cytokine response induced by infection[29,63].

Additional opportunities for the further development and medical translation of immune-like designer cell-based therapies like ALICE include clinical manipulation of patient-derived cells, non-invasive monitoring of cell stability, and the potential for long-duration performance stretching over many months or longer[64]. These challenges do not seem insurmountable. One technical aspect that should promote such efforts would be the modification of implantation materials, which previous efforts have shown to increase the safety and efficacy of cellular transplants and thereby facilitate the maintenance of the engineered cellular functions for extended periods of time in vivo[65].

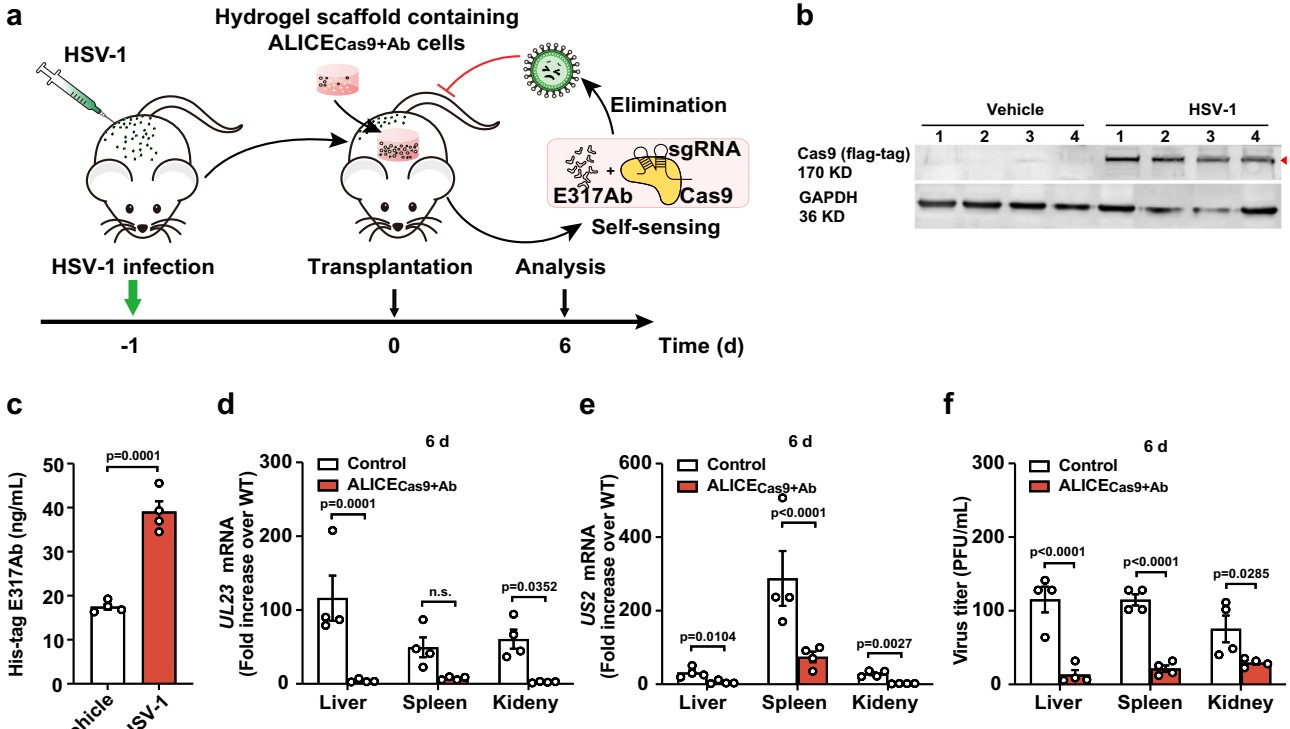

**Fig. 6 | Sense-and-destroy against HSV-1 mediated by ALICE$_{Cas9+Ab}$ in a virus-infected mouse model. a** Schematic illustration of ALICE$_{Cas9+Ab}$ for autonomous sense-and-destroy against HSV-1 in a virus-infected mouse model. Three sgRNAs (pYW102/pYW172/pYW188, respectively targeting *US8/UL29/UL52*)-transgenic HEK$_{ALICE-Cas9-E317Ab}$ cells (ALICE$_{Cas9+Ab}$) or unmodified HEK-293T cells (Control) were encapsulated into hydrogel-based scaffolds. At 20 hpi of HSV-1 ($2 \times 10^7$ PFU) by intraperitoneal injection, mice were implanted with the ALICE$_{Cas9+Ab}$ by intraperitoneal surgery. The antiviral effects of ALICE$_{Cas9+Ab}$ in mice were evaluated by detecting viral titers in indicated mouse organs (liver, spleen, kidney) at 6 days post-transplantation. **b** Western blot analysis of HSV-1-inducible Cas9 expression in hydrogel-based scaffolds and **c** validation of HSV-1-inducible E317Ab expression. HSV-1-infected mice ($2 \times 10^7$ PFU, HSV-1 group) or uninfected (0, Vehicle group) were all transplanted with hydrogel-based scaffolds containing pYW102/pYW172/pYW188-transgenic HEK$_{ALICE-Cas9-E317Ab}$ cells. Total protein from hydrogel implants isolated from mice were extracted for Western blot analysis at 6 days post-transplantation. The red arrowhead indicates the expected Cas9 band. E317Ab levels in the blood were analyzed by using a His-tag ELISA at 6 days post-transplantation. **d**, **e** qPCR assay of HSV-1 *UL23/US2* mRNA in liver/spleen/kidney at 6 days post-transplantation between Control and ALICE$_{Cas9+Ab}$ group, using primers listed in Supplementary Table 2. The relative expression values were calculated using the ΔΔct method based on the expression levels of the viral genes *UL23/US2* in organs of WT mice without HSV-1 infection and intraperitoneal transplantation of any hydrogel. **f** Viral titers in mice. The mice were processed as described in (**a**). Virus in tissues, such as liver, spleen, kidney, were measured titrated at 6 days post-transplantation. IgG levels in the blood were analyzed by using an IgG ELISA at 6 days post-transplantation. Number 1–4 represents four indenpent mice in b; Data in c-f are expressed as means ± SEM; *P* values in **c** were calculated by two-tailed unpaired *t*-test; *P* values in **d**–**f** were calculated by two-way ANOVA with Bonferroni's post hoc test; *n* = 4 mice. Source data are provided as a Source Data file.

Collectively, the modular design of ALICE systems, comprising a selected host cell chassis, a pathogen-detection sensor module, a rewired endogenous signaling pathway, diversely regulated output pathways, and delivered by designer cells or AAV-vectors, should flexibly accommodate the specific application requirements (suitable antigen, suitable control agents) for many infectious diseases. We anticipate ALICE$_{im}$ can function as an artificial innate immune system, protecting the host organism through nonspecific immune defense by the induction of interferons. We foresee that the diverse examples of ALICE systems we have demonstrated in the present study can serve as illustrative models, which can be readily adapted for the development of immune-like designer cells to achieve sense-and-destroy functions for a potentially very large number of pathogens that target mammals.

## Methods
### Ethical statement
All the experiments involving mice were performed according to the approved protocols by the East China Normal University (ECNU) Animal Care and Use Committee (protocol ID: m20180403). All the procedures for sample or data collection used were carried out in compliance with the Ministry of Science and Technology of the People's Republic of China on Animal Care Guidelines. All mice were euthanized after the experiments.

### Cloning and vector construction
Design and construction details of each plasmid are provided in Supplementary Table 1. All genetic components were confirmed by DNA sequencing (Genewiz Inc.).

### Cell culture
Human cervical adenocarcinoma cells (HeLa, ATCC: CCL-2), HEK-293T-derived Hana3A cells engineered for the stable expression of G$_{\alpha o \lambda \Phi}$ and chaperones RTP1/RTP2/REEP1, HEK-293T-derived HEK-293A cells containing a stably integrated copy of the E1 gene (ThermoFisher, cat. no. R70507), telomerase-immortalized human mesenchymal stem cells (hMSC-TERT, ATCC: SCRC4000), human embryonic kidney cells (HEK-293T, ATCC: CRL-11268), African green monkey kidney epithelium-derived Vero cells (Vero, ATCC: CCL-81), Vero E6 cells (ATCC: CRL-1586) rhadomyosarcoma (RD, ATCC: CCL-136), Huh7.5.1 cells and Huh7-NTCP cells were cultivated in DMEM (Gibco, cat. no. 31600-083) supplemented with 10% (v/v) fetal bovine serum (FBS, Biological Industries, cat. no. 04-001-1C) and 1% (v/v) penicillin/streptomycin solution (Sangon Biotech, cat. no. B540732-0010). Aedes albopictus cells (C6/36,

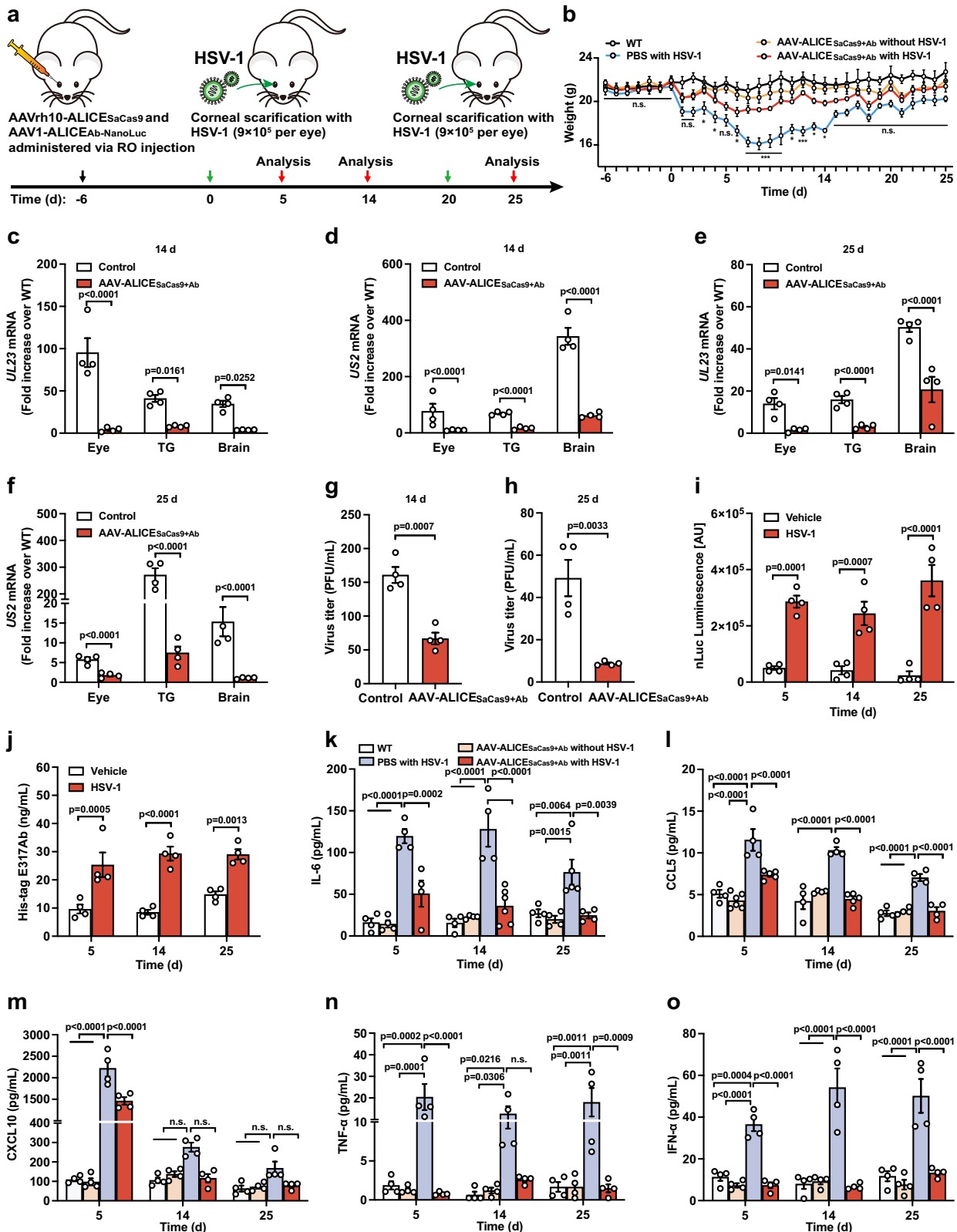

ATCC: CRL-1660) were incubated at 28 °C, and all other cell types were incubated at 37 °C in a humidified atmosphere incubator, containing 5% $CO_2$ and were regularly tested for the absence of mycoplasma and bacterial contamination.

## Virus preparation

HSV-1 and replication-competent HSV-1 (strain 17) with non-necessary gene (*UL2*) replaced by enhanced green fluorescent protein gene (EGFP-labeled HSV-1) were gifted by Professor Ping Wang (Tongji University) and Professor Erguang Li (Nanjing University), respectively. HSV-1 was propagated in Vero cells. H1N1, Pteropine orthoreovirus PRV2P, hCoV-229E and SARS-CoV-2 isolate BetaCoV/Singapore/2/2020 (GISAID accession no. EPI_ISL_406973) were propagated in Vero E6 cells. EV-A71 isolate 5865/SIN/000009 (GenBank accession no. AF316321) was produced using HeLa cells. VSV was gifted by Professor Peng Zhou (Wuhan Institute of Virology, Chinese Academy of

**Fig. 7 | Long-term sense-and-destroy against HSV-1 mediated by AAV-ALICE$_{SaCas9+Ab}$ in a herpetic simplex keratitis mouse model. a** Schematic illustration of AAV-ALICE$_{SaCas9+Ab}$ for autonomous sense-and-destroy against HSV-1 in a herpetic simplex keratitis mouse model. Two packaged AAV vectors, AAVrh10-ALICE$_{SaCas9}$ ($5 \times 10^{11}$ PFU) and AAV1-ALICE$_{Ab}$ ($5 \times 10^{11}$ PFU), were simultaneously injected into each mouse via RO route 6 days prior to initial HSV-1 infection. Control mice were injected with PBS. At 0 day and 20 days, HSV-1 ($9 \times 10^5$ PFU) or PBS was delivered by corneal scarification of each eye using a 28-gauge needle. The antiviral effects of ALICE-packaged AAVs in mice was evaluated by measuring residual viral titers in the indicated mouse organs (eye, TG, brain) at 14- and 25-days post initial HSV-1 infection. **b** Change in body weight. *P* values are provided in the Source data file. **c** qPCR analysis of HSV-1 *UL23/US2* mRNA in mice at 14 days (**c, d**) and 25 days (**e, f**) post initial HSV-1 infection. Viral titers in mice. Virus in the brain was titrated at 14 days (**g**) and 25 days (**h**) post initial HSV-1 infection. **i** Assay of HSV-1-induced nanoLuc expression in mice. NanoLuc levels from AAV-ALICE$_{SaCas9+Ab}$-treated mice infected with or without HSV-1, were analyzed at day 5, 14 and 25 post initial HSV-1 infection. **j** Assay of HSV-1-inducible E317Ab expression in mice. Assay of cytokine expression in mice. Cytokine levels, IL-6 (**k**), CCL5 (**l**), TNF-α (**m**), IFN-α (**n**), and CXCL10 (**o**) in the blood were analyzed by flow cytometry at day 5, 14, and 25 post initial HSV-1 infection. White bars (WT), brown bars (AAV-ALICE$_{SaCas9+Ab}$ without HSV-1), blue-gray bars (PBS with HSV-1), red bars (AAV-ALICE$_{SaCas9+Ab}$ with HSV-1) in (**k–o**). Data in **c–f** are normalized to wild-type mice (WT); Data in **b–o** are expressed as means ± SEM; *P* values in **c–f, i–o** were calculated by two-way ANOVA with Bonferroni's post hoc test; *P* values in **g, h** were calculated by two-tailed unpaired *t*-test; *n* = 4–9 mice in (**b**), *n* = 4 mice in (**c–j**), *n* = 4–6 mice in (**k–o**). n.s. not significant. Source data are provided as a Source Data file.

Sciences). Lentivirus, AAV and Adenovirus 5 (ADV) were purchased from ObiO Technology (Shanghai) Corp., Ltd. DENV-2 strain was propagated in *Aedes albopictus* cells at 28 °C.

## Virus titration
HSV-1 infectivity was evaluated using 50% tissue culture infective dose assays (TCID$_{50}$). Briefly, cells seeded in 96-well plates were infected with serially diluted virus, eight replicates per dilution. For each dilution, the number of wells that were positive for cytopathic effect (CPE) was scored. A Reed and Muench calculation was then performed to determine the 50% infectious dose[66]. And CPE was assessed at 4–7 dpi. HSV-1/EGFP-labeled HSV-1 was titrated by plaque assay in Vero cells. H1N1, PRV2P, hCoV-229E and SARS-CoV-2 were titrated by plaque assay in Vero E6 cells. EV-A71 was titrated on RD cells. The quantification of viruses (SARS-CoV-2, HCV, HBV, ADV, HSV-1) was performed by qPCR using the primers target corresponding genes [receptor binding domain for SARS-CoV-2 Spike protein (*RBD*), HCV genomic RNA, HBV pre-genomic RNA (pgRNA), *E1B/E2 early*, *UL23/UL30/US2*] and QuantiFast SYBR Green RT-PCR kit (Qiagen) following the manufacturer's instructions. All primers are listed in Supplementary Table 2.

## Cell transfection and virus infection
All cell lines, except Huh7.5.1 and Huh7-NTCP cells, were transfected with an optimized polyethyleneimine (PolyScience)-based protocol. Huh7.5.1 cells/Huh7-NTCP cells were transfected using lipofectamine 2000 (Invitrogen) according to the manufacturer's instructions.

Transgenic cells were infected with different viruses (H1N1, EV-A71, PRV2P, VSV, Lentivirus, AAV, DENV-2, SARS-CoV-2, hCoV-229E, ADV, and HSV-1) for 1 h, and transgenic-Huh7.5.1 cells/Huh7-NTCP cells were incubated with HCV/HBV for 24 h. SEAP production in culture supernatants was quantified at indicated time points by SEAP reporter assay as previously reported[67]. SEAP expression levels in the cell culture supernatant were quantified using a Synergy H1 hybrid multi-mode microplate reader with Gen5 software (version: 2.04).

## Sense-and-clearance against multiple viruses mediated by ALICE$_{im}$ in mammalian cells
Virus-inducible cytokines production in ALICE$_{im}$. pYW274/pYW365-transgenic cells were incubated with HCV (MOI = 3), HBV [1000 virion genome equivalents (vge)/cell], ADV (MOI = 10), HSV-1 (MOI = 5) and Vehicle (equal volume of DMEM) as control for 24 h. The IFN-α/IFN-β production in culture supernatants was quantified by ELISA at 2/4 dpi, using a human IFN-α ELISA kit (Beyotime, cat. no. PI505) or human IFN-β ELISA kit (Beyotime, cat. no. PI572) according to manufacturer's instructions.

qPCR analysis of the viral transcript genes in ALICE$_{im}$. pYW274/pWS67-transgenic cells (ALICE$_{sen}$-SEAP), pYW274/pYW365-transgenic cells (ALICE$_{im}$-IFNα) or pYW274/pYW327-transgenic cells (ALICE$_{im}$-IFNβ) were incubated with the HCV (MOI = 3), HBV (1000 vge/cell), ADV (MOI = 10), HSV-1 (MOI = 5) for 24 h. The relative viral mRNA expression was quantified by qPCR at 2/4 dpi. All data were normalized to the viral gene expression levels in ALICE$_{sen}$-SEAP control group infected with the corresponding viruses. All primers used are provided in Supplementary Table 2.

## HSV-1 infection and inhibition assay
EGFP-labeled HSV-1 infection and inhibition in cells were assessed by measuring EGFP fluorescence intensity using Synergy™ H4 Hybrid Multi-Mode Microplate Reader (BioTek Instruments Inc.) with an excitation wavelength of 479 nm and an emission wavelength of 525 nm. The relative fluorescence intensity value is the ratio of the fluorescence intensity of treated cells and untreated cells subtracting the blank (media only) fluorescence intensity.

EGFP-labeled HSV-1 infection and inhibition in mice were examined in the liver, spleen, and kidney tissues. Briefly, indicated tissues of mice were isolated, collected, and washed in cold PBS three times. Tissues were then cut into small pieces, kept on ice and transferred to a homogenizer. Half the tissue in RNAiso Plus kit (Takara Bio, cat. no. 9108) was vigorously vortexed, homogenized and centrifuged ($12,000 \times g$, 10 min) at 4 °C for RNA extraction and qPCR assay. Half the tissue in sterile PBS was vigorously vortexed, homogenized and centrifuged as above for titration[68].

## Sense-and-deletion against viruses mediated by ALICE$_{Cas9}$
The ALICE$_{Cas9}$ device was loaded into HEK293T cells by transfecting pYW274, pYW169, and pYW444 sgRNAs targeting both ADV and HSV-1 genomic DNA 24 h prior to virus incubation. ALICE$_{Cas9}$ cells were incubated with single virus (ADV, MOI = 10; or HSV-1, MOI = 5) or double viruses (simultaneous infection of ADV and HSV-1, MOI = 10 or 5, respectively) for 3 h. The relative viral mRNA expression of *E1B/E2 early* genes from ADV, and *UL23/UL30* genes from HSV-1 was quantified by qPCR at 2 dpi. All data were normalized to the viral gene expression levels in non-ALICE$_{Cas9}$ cells where HEK-293T cells were co-transfected with pYW274/pWS67/pcDNA3.1 and infected with a single virus (ADV, MOI = 10; or HSV-1, MOI = 5) for 3 h. All primers used are provided in Supplementary Table 2.

## Stable cell lines construction
The monoclonal HEK$_{ALICE-SEAP-Cas9}$ cell line, stably transgenic for HSV-1-inducible SEAP and Cas9 expression, was constructed by co-transfecting HEK-293T ($1 \times 10^5$ cells) with 400 ng pYW306 (ITR-P$_{ALICE6}$-SEAP-P2A-Cas9-pA::P$_{mPGK}$-puromycin-E2A-STING-PEST-pA-ITR) and 20 ng of the Sleeping Beauty transposase expression vector (P$_{hCMV}$-SB100X-pA), followed by selection in culture medium containing 1 μg/mL puromycin (Thermo Fisher Scientific, cat. no. A1113803) for 10 days. The surviving population was selected for further cultivation and stimulated with HSV-1 (MOI = 0 or 5). Monoclonal cell lines with optimal HSV-1-inducible SEAP and Cas9 production was selected for follow-up studies. Meanwhile, monoclonal cell lines with only optimal HSV-1-inducible SEAP production were selected as HEK$_{ALICE-SEAP}$ cell line for follow-up studies.

The monoclonal HEK$_{ALICE-Cas9-E317Ab}$ cell line, stably transgenic for HSV-1-inducible Cas9 and E317Ab expression, was constructed by transfecting HEK$_{ALICE-SEAP-Cas9}$ ($7 \times 10^4$ cells) with 200 ng pYW383 (P$_{ALICE6}$-E317Ab-6×His-P2A-mCherry-pA::P$_{mPGK}$-Zeocin-pA), then selected by 1 µg/mL puromycin and 100 µg/mL Zeocin (Thermo Fisher Scientific, cat. no. R25001) for 10 days. The monoclonal cell lines with optimal HSV-1-inducible E317Ab production were selected for follow-up studies.

The monoclonal HEK$_{ALICE-Cas9-sgRNAs-E317Ab}$ cell line, stably transgenic for HSV-1-inducible Cas9 and E317Ab expression, and a constitutive expression of HSV-1-targeted sgRNA$_{UL29}$, sgRNA$_{UL52}$ and sgRNA$_{US8}$, was constructed by transfecting HEK$_{ALICE-Cas9-E317Ab}$ ($7 \times 10^4$ cells) with 200 ng pYW412 (P$_{U6}$-sgRNA$_{UL29}$-sgRNA$_{UL52}$-sgRNA$_{US8}$-P$_{SV40}$-BSD-pA), then selected by 5 µg/mL blasticidin (Thermo Fisher Scientific, cat. no. A1113903) for 10 days. The monoclonal cell lines with optimal antiviral effects were selected and mixed for follow-up studies.

## qPCR assay

Total RNA in treated HEK-293 cells or in liver, spleen, kidney tissues of treated mice were isolated using the RNAiso Plus kit (Takara Bio, cat. no. 9108). The purified RNA (1 µg) was reversely transcribed into cDNA using PrimeScript™ Reverse Transcription Kit with the gDNA Eraser (Takara Bio, cat. no. RR047). qPCR was performed on LightCycler® 96 instrument (Roche) using SYBR *Premix Ex Taq*™ (Takara Bio, cat. no. RR420) with special primers listed in Supplementary Table 2. The following amplification parameters were used: 95 °C for 10 min, 40 cycles of 95 °C for 30 s, 58 °C for 30 s, and a final cooling at 37 °C for 30 s. The expression of human/mouse housekeeping gene *glyceraldehyde 3-phosphate dehydrogenase* (*GAPDH*) was used as an internal control. The fold change for relative mRNA level was evaluated using the comparative ΔΔCt method[67].

## T7 endonuclease 1 (T7E1) mismatch detection assay

The deletion efficiency by HSV-1-inducied Cas9 in HEK-293T cells were measured using the primers listed in Supplementary Table 3 as reported[67]. Total genomic DNA of cells was extracted by a TIANamp genomic DNA kit (Tiangen bio, cat. no. DP304) according to the manufacturer's protocol. sgRNA-targeted *CCR5* gene were PCR-amplified from total genomic DNA using the 2×Taq Plus Master Mix II (Dye Plus) DNA polymerase (Vazyme Inc, cat. no. P213) with the primers listed in Supplementary Table 3. PCR amplicons were purified by a universal DNA purification kit (Tiangen bio, cat. no. DP214). Total 20 µL mixture containing 500 ng purified PCR production and 2 µL 10×M buffer (Takara Bio, cat. no. 1060S) was re-annealed (95 °C for 5 min, then decreased to room temperature) to form heteroduplex DNA. The heteroduplexed DNA was digested using 0.5 µL of T7 endonuclease I (New England BioLabs, cat. no. M0302) and incubated for 2 h at 37 °C. Digested products were separated on a 2% agarose gel. The deletion efficiency by HSV-1-inducied Cas9 was calculated with the following formula: deletion efficiency = $100\% \times b/(a+b)$, where a represents the intensity of undigested PCR amplicons and b represents the intensities of the T7E1-digested products.

## Plaque assay

Transfection of pYW274/pYW169 with an HSV-1-targeting sgRNA$_{US8}$/sgRNA$_{UL29}$/sgRNA$_{UL52}$ (pYW102/pYW172/pYW188) or a nonsense sgRNA (pWS68) in HEK-293T cells was performed 20 h prior to EGFP-labeled HSV-1 infection (MOI = 5) for 3 h. After virus incubation for 3 h, supernatant was removed and placed with fresh media containing 1% sterile low melting point agarose (Yeasen Biotech, cat. no.10214ES08). At 48 hpi, solidified agarose medium was removed and HSV-1-infected cells were fixed in 10% trichloroacetic acid (Aladdin, cat. no. 76-03-9) and stained with 0.05% crystal violet (Aladdin, cat. no. 548-62-9). Cells were washed three times in sterile 1×PBS (Sangon Biotech, cat. no. A610100). Micrograph profiling cell activity was performed by microscopy.

## Western blot analysis

Cells infected with or without HSV-1 or cells in the hydrogel-based scaffold isolated from mice were harvested with a RIPA lysis buffer (Yeasen Biotech, cat. no. 20101ES60). The lysates were centrifuged at $10,000 \times g$ for 15 min at 4 °C and the protein concentrations were quantified by BCA protein quantification kit (Yeasen Biotech, cat. no. 20201ES76). Proteins samples were loaded on a SDS-PAGE and then electrotransferred onto a polyvinylidene difluoride membrane (PVDF, Millipore, cat. no. ISEQ00010). The membrane was blocked with 5% non-fat milk and incubated with primary antibodies [monoclonal rabbit anti-cGAS (CST, cat. no. 15102T, clone no. D1D3G, 1:1000), monoclonal rabbit anti-STING (CST, cat. no. 13647S, clone no. D2P2F, 1:1000), monoclonal mouse anti-β-Tubulin (Yesen, cat. no. 30301ES40, 1:1000), monoclonal mouse anti-flag (Abcam, cat. no. ab125243, clone no. FG4R, 1:1000), monoclonal rabbit anti-GAPDH (Yesen, cat. no. 30202ES40, 1:2000)]. The membrane was washed 3 times in TBS with 0.05% tween 20 (Sangon biotech, cat. no. 9005-64-5). Immune complexes were detected using secondary antibodies [Alexa fluor-based Goat Anti-Mouse IgG (H + L) (Yesen, cat. no. 33219ES60, 1:25,000), Alexa fluor-based Goat Anti-Rabbit IgG (H + L) (Yesen, cat. no. 33119ES60, 1:25,000)]. Images were scanned using Alpha Innotech Flour Chem-FC2 imaging system (San Leandro)[69].

## ELISA assay

Expression levels of E317Ab containing a His-tag in culture supernatants or mouse serum were measured using a His-tag ELISA detection kit (GenScript, cat. no. L00436). Mouse serum levels of IL-6, tumor necrosis factor (TNF)-α, interferon (IFN)-γ, and IgG were detected using the LEGEND Max™ mouse IL-6 ELISA kit (BioLegend, cat. no. 431307), mouse TNF-α ELISA kit (MultiSciences, cat. no. 70-EK282/3-96), mouse IFN-gamma ELISA kit (MultiSciences, cat. no. 70-EK280/3-96), and the mouse IgG ELISA kit (MultiSciences, cat. no. 70-EK271-96) according to manufacturer's instructions.

## NanoLuc assay

NanoLuc levels in cell culture or plasma were measured using the Promega Nano-Glo™ Luciferase Assay System (Promega, cat.no. N1110) according to the manufacturer's instructions. All assay components (reagents and samples) were equilibrated to room temperature prior to use. Nano-Glo™ Luciferase Assay Reagent was prepared by combining one volume of Nano-Glo™ Luciferase Assay Substrate with 50 volumes of Nano-Glo™ Luciferase Assay Buffer. A volume of reagent equal to that of sample was added to each well. After a 3 min incubation, luminescence was measured using the Synergy™ H4 Hybrid Multi-Mode Microplate Reader (BioTek Instruments Inc.)

## Fluorescence imaging

EGFP expression was measured using an Olympus inverted fluorescence microscope (Olympus IX71, TH4-200) equipped with an Olympus digital camera, a pE-100-LED (CoolLED) as the transmission light source, a Spectra X (Lumencor) as the fluorescent light source, a 10× objective, a 488 nm/509 nm (B/G/R) excitation/emission filter set, and Image-Pro Express C software (version ipp6.0).

## Cytokine and whole blood detection

Mouse serum levels of cytokines, including IL-6, CCL5, CXCL10, TNF-α, and IFN-α, were detected using LEGENDplex™ Multi-Analyte Flow Assay Kit (BioLegend, cat.no. 740625) according to manufacturer's instructions. The samples were acquired on a BD LSRFortessa™ Flow Cytometer (BD Biosciences) applying a 488 nm laser with 536/40 (BP) filter for the PE fluorochrome and a 638 nm laser with 675/20 (BP) for the APC fluorochrome. The results were analyzed using LEGENDplex™ Data Analysis Software V.8.0 (Vigene Tech Inc., USA). The concentration of each growth factor/cytokine was determined by means of a standard curve generated during the performance of the assay.

The levels of white blood cells, lymphocytes cells, and monocytes cells in mouse blood were measured according to the manufacturer's protocol by Shanghai Model Organisms Inc.

## Transwell®-based assay

Prevention of virus spread. $5 \times 10^4$ immune-like designer cells (ALICE$_{Cas9}$/ALICE$_{Ab}$/ALICE$_{Cas9+Ab}$) or HEK-293T cells (Control) per well incubated with EGFP-labeled HSV-1 (MOI = 5) for 3 h were then separately seeded on transwell polycarbonate membrane inner chambers with an 8 μm pore size (Corning, cat. no. 3428), and then cocultured with HEK-293T cells ($5 \times 10^4$ cells/well) cultured for 18 h on transwell outer chambers. After 48 h of co-incubation, the fluorescence intensity of HEK-293T cells on outer chambers was detected by the Synergy™ H4 Hybrid Multi-Mode Microplate Reader (BioTek Instruments Inc.).

Prevention of virus infection. HEK-293T cells ($5 \times 10^4$ cells/well) seeded on the transwell outer chambers, cultured for 18 h, and then infected with EGFP-labeled HSV-1 (MOI = 5) for 3 h. After adding fresh media, the infected cells were cocultured with immune-like designer cells (ALICE$_{Cas9}$/ALICE$_{Ab}$/ALICE$_{Cas9+Ab}$) or HEK-293T cells (Control) ($5 \times 10^4$ cells/well) seeded on the transwell polycarbonate membrane inner chambers with an 8-μm pore size (Corning, cat. no. 3428). After 48 h of co-incubation, the fluorescence intensity of individual cells on the transwell polycarbonate membrane inner chambers was detected by the Synergy™ H4 Hybrid Multi-Mode Microplate Reader (BioTek Instruments Inc.).

## Sense-and-destroy against HSV-1 mediated by ALICE in mice

To test the HSV-1-inducible transgene expression in mice, $4 \times 10^6$ HEK$_{ALICE-Cas9-E317Ab}$ or HEK$_{ALICE-SEAP-Cas9}$ cells were co-transfected with three sgRNAs targeting *US8* (pYW102, 4 μg, P$_{U6}$-sgRNA$_{US8}$), *UL29* (pYW172, 4 μg, P$_{U6}$-sgRNA$_{UL29}$) and *UL52* (pYW188, 4 μg, P$_{U6}$-sgRNA$_{UL52}$), while control cells ($4 \times 10^6$ HEK$_{ALICE-Cas9-E317Ab}$) were transfected with pcDNA3.1 (12 μg, P$_{hCMV}$-MCS-pA). Wild-type HEK-293T ($4 \times 10^6$ cells) were transfected with pcDNA3.1 (12 μg). $4 \times 10^6$ engineered cells for each mouse were encapsulated into a cylindrical hydrogel-based scaffolds using 300 μL hydrogel scaffold solution (Sigma-Aldrich, cat. no. HYS020).

The female BALB/c wild-type mice (4-week-old; ECNU Laboratory Animal Center) were kept in the animal house maintained at $22 \pm 2\,°C$, with a 12 h light–dark cycle and free access to food and water. To test the autonomous sense-and-destroy against virus of immune-like designer cells (ALICE$_{Cas9}$/ALICE$_{Ab}$/ALICE$_{Cas9+Ab}$) in mice, wild-type female BALB/c mice (four-week-old, ECNU Laboratory Animal Center) were randomly divided into six groups including WT group, sham operation group, control group, ALICE$_{Cas9}$ group, ALICE$_{Ab}$ group, and ALICE$_{Cas9+Ab}$ group: (1) Wild-type BALB/c mice without any treatment were used as negative control (WT); (2) Wild-type BALB/c mice were intraperitoneally implanted with hydrogels containing pYW102/pYW172/pYW188-cotransfected HEK$_{ALICE-Cas9-E317Ab}$ cells without subsequent HSV-1 infection, marked as sham operation group; (3) Control mice were implanted with hydrogels containing pcDNA3.1-transfected wild-type HEK-293T cells; (4) Wild-type BALB/c mice were intraperitoneally implanted with hydrogels containing pYW102/pYW172/pYW188-cotransfected HEK$_{ALICE-SEAP-Cas9}$ cells (ALICE$_{Cas9}$); (5) pcDNA3.1-transfected HEK$_{ALICE-Cas9-E317Ab}$ cells (ALICE$_{Ab}$); and 6) pYW102/pYW172/pYW188-cotransfected HEK$_{ALICE-Cas9-E317Ab}$ cells (ALICE$_{Cas9+Ab}$). One day after implantation, control, ALICE$_{Cas9}$, ALICE$_{Ab}$, and ALICE$_{Cas9+Ab}$ groups were intraperitoneally injected with EGFP-labeled HSV-1 ($2 \times 10^7$ PFU/mL, 200 μL per mouse) and the sham operation group were intraperitoneally injected with PBS solution (200 μL per mouse).

Mice were retro-orbitally bled at 2, 4, and 6 days post-HSV-1 injection or 30 days post-transplantation. Serum was separated from whole blood by centrifugation at $5000 \times g$ for 10 min. Expression levels of E317Ab containing a His-tag in blood were quantified using an his-tag ELISA (GenScript, cat. no. L00436). At 2, 4, and 6 days post-HSV-1 injection or 30 days post-transplantation, mice were euthanized, and the organs (liver, spleen, and kidney) were excised. Virus in tissues was evaluated by titration and qPCR using primers listed in Supplementary Table 2.

## Inhibition of HSV-1 transmission mediated by ALICE$_{Cas9+Ab}$ in mice

To test the ALICE$_{Cas9+Ab}$ mediated inhibition of HSV-1 transmission in mice, HEK$_{ALICE-Cas9-E317Ab}$ cells ($4 \times 10^6$ cells) co-transfected with three sgRNAs targeting *US8/UL29/UL52* (pYW102/pYW172/pYW188, 4 μg, respectively) were incubated with or without EGFP-labeled HSV-1 (MOI = 1) for 3 h. HEK-293T control cells ($4 \times 10^6$ cells) transfected with 12 μg pcDNA3.1 were incubated with EGFP-labeled HSV-1 (MOI = 1) for 3 h. $4 \times 10^6$ cells of each mouse were encapsulated into a cylinder of hydrogel-based scaffolds using 300 μL hydrogel scaffold solution as described above.

The female BALB/c wild-type mice (4-week-old; ECNU Laboratory Animal Center) were kept in the animal house maintained at $22 \pm 2\,°C$, with a 12 h light-dark cycle and free access to food and water. To test the inhibition of HSV-1 transmission in mice, wild-type female BALB/c mice were randomly divided into four groups: WT, sham operation, control, and ALICE$_{Cas9+Ab}$ groups. (1) Wild-type BALB/c mice without any treatment were used as WT group. (2) Wild-type BALB/c mice intraperitoneally implanted with hydrogels containing pYW102/pYW172/pYW188-co-transfected HEK$_{ALICE-Cas9-E317Ab}$ cells without HSV-1 infection, were marked as sham operation group. (3) Control mice were implanted with hydrogels containing pcDNA3.1-transfected HEK-293T cells with HSV-1 infection. 4) Wild-type BALB/c mice intraperitoneally implanted with hydrogels containing pYW102/pYW172/pYW188-cotransfected HEK$_{ALICE-Cas9-E317Ab}$ cells with HSV-1 infection were marked as ALICE$_{Cas9+Ab}$ group. IL-6 or E317Ab levels in serum were quantified by ELISA. At 2, 4, and 6 days post-transplantation, mice were euthanized, and the organs (liver, spleen, and kidney) were excised. The viral infection and inhibition efficacy in tissues were performed as described in "Sense-and-destroy against HSV-1 mediated by ALICE in mice".

## Sense-and-destroy against HSV-1 mediated by ALICE$_{Cas9+Ab}$ in a virus-infected mouse model

To establish an HSV-1-infected mouse model, wild-type female BALB/c mice were randomly divided into four groups including WT group, sham operation group, control group, and ALICE$_{Cas9+Ab}$ group. Mice of control group and ALICE$_{Cas9+Ab}$ group were intraperitoneally injected with HSV-1 ($2 \times 10^7$ PFU/mL, 200 μL per mouse). HEK$_{ALICE-Cas9-E317Ab}$ cells were co-transfected with three sgRNAs targeting *US8/UL29/UL52* (pYW102/pYW172/pYW188, 4 μg, respectively). Wild-type HEK-293T ($4 \times 10^6$ cells) were transfected with pcDNA3.1 (12 μg). Each mouse implanted with indicated $4 \times 10^6$ cells were encapsulated into a cylindrical hydrogel-based scaffolds using 300 μL hydrogel scaffold solution. The female BALB/c wild-type mice (4-week-old; ECNU Laboratory Animal Center) were kept in the animal house maintained at $22 \pm 2\,°C$, with a 12 h light–dark cycle and free access to food and water.

To test the autonomous sense-and-destroy of virus mediated by ALICE$_{Cas9+Ab}$, HSV-1-infected mice were intraperitoneally implanted with hydrogels containing pYW102/pYW172/pYW188-cotransfected HEK$_{ALICE-Cas9-E317Ab}$ cells (ALICE$_{Cas9+Ab}$ group) or pcDNA3.1-transfected HEK-293T cells (control group). Wild-type BALB/c mice were implanted with hydrogels containing ALICE$_{Cas9+Ab}$ cells (sham-operation group). Wild-type BALB/c mice without any treatment were used as negative control (WT group). Serum was collected as described above and E317Ab or IgG levels were quantified by ELISA. At 6 days post-transplantation, mice were euthanized, and the organs (liver, spleen, and kidney) were excised. The viral infection and inhibition efficacy in tissues were examined as described above.

## The herpetic simplex keratitis mouse model

For AAV inoculation, mice anesthetized with ketamine/xylazine were administered the two indicated AAV vectors by RO injection. The female BALB/c wild-type mice (4-week-old; ECNU Laboratory Animal Center) were kept in the animal house maintained at $22 \pm 2$ °C, with a 12 h light–dark cycle and free access to food and water. AAV vectors marked as AAV-ALICE$_{SaCas9+Ab}$ consisted of two packaged AAV vectors. AAVrh10-ALICE$_{SaCas9}$ carrying an HSV-1-induced SaCas9 and a constitutive expression of HSV-1-targeted *ICP4* sgRNA (pYWG4, ITR-P$_{ALICE6}$-SaCas9-pA::P$_{U6}$-sgRNA$_{ICP4}$-ITR; $5 \times 10^{11}$ PFU per mouse) and AAV1-ALICE$_{Ab}$ carrying an HSV-1-induced E317Ab-P2A-nanoLuc and a constitutive expression of STING-PEST (pYW414, ITR-P$_{ALICE6}$-E317Ab-6×His-P2A-NanoLuc-pA::P$_{hCMV}$-STING-PEST-pA-ITR; $5 \times 10^{11}$ PFU per mouse), were simultaneously injected into each mouse (total 100 μL per mouse, 50 μL per eye) via RO injection. All AAVs were packaged by Shanghai Taitool Bioscience Co. Ltd. Equal volumes (total 100 μL per mouse, 50 μL per eye) of PBS solution were injected via RO injection as the control group. The eye, TG, and brain were collected at the indicated time points for further analysis.

For ocular HSV-1 infection, wild-type female BALB/c mice (six-week-old, ECNU Laboratory Animal Center) were anesthetized by intraperitoneal injection of ketamine (100 mg/kg) and xylazine (12 mg/kg) and infected with EGFP-labeled HSV-1 ($9 \times 10^5$ PFU) following corneal scarification of each eye using a 28-gauge needle.

To test the autonomous sense-and-destroy of HSV-1 mediated by AAV-ALICE$_{SaCas9+Ab}$ in herpetic keratitis mouse model, six-week-old wild-type female BALB/c mice were randomly divided into four groups including WT group, sham operation group, control group, and AAV-ALICE$_{SaCas9+Ab}$ group. Mice were simultaneously RO injection with two packaged AAV vectors (AAVrh10-ALICE$_{SaCas9}$ and AAV1-ALICE$_{Ab}$) 6 days prior to initial HSV-1 infection. Meanwhile, control mice were injected with PBS. At 0 day and 20 days, mice containing AAV-ALICE$_{SaCas9+Ab}$ system were administered with corneal HSV-1 infection (marked as AAV-ALICE$_{SaCas9+Ab}$ group) or non-infection (marked as sham-operation group). Control mice were administered with corneal HSV-1 infection at 0 day and 20 days (marked as control group). Wild-type BALB/c mice without any treatment were used as WT group. Mouse blood was RO collected at 5, 14, and 25 days post-initial HSV-1-infection, and serum was separated as described above. NanoLuc, E317Ab, and IgG levels in serum were quantified using the corresponding detection kits according to manufacturer's instructions. At 14 and 25-days post-initial HSV-1 infection, mice were euthanized and the organs (eye, TG, and brain) were excised. The viral infection and inhibition efficacy in tissues were evaluated as described in "Sense-and-destroy against HSV-1 mediated by ALICE in mice".

## Statistical analysis

All in vitro data are expressed as mean ± SD of three independent experiments ($n = 3$ or 4). All micrographs were repeated by three independent experiments with similar results. For the animal experiments, each treatment group consisted of randomly selected mice ($n = 4$ to 9). Neither animals nor samples were excluded from the study. Comparisons between groups were performed using two-tailed Student's $t$ test as means ± SEM. Comparison of the data from multiple groups against one group was performed using a one-way analysis of variance (ANOVA) followed by a Dunnett's post hoc test or a two-way ANOVA with Bonferroni's post hoc test. GraphPad Prism software (version 8.3) was used for statistical analysis. $P$ values <0.05 were considered statistically significant.

## Reporting summary

Further information on research design is available in the Nature Portfolio Reporting Summary linked to this article.

## Data availability

All data associated with this study are present in the manuscript or the Supplementary Materials. All genetic components related to this paper are available with a material transfer agreement and can be requested from H.Y. (hfye@bio.ecnu.edu.cn) or L.-F.W. (linfa.wang@duke-nus.edu.sg). Source data are provided with this paper.

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

## Acknowledgements

We thank Professor Erguang Li (Nanjing University) for gifting us the EGFP-labeled HSV-1 strain and Professor Ping Wang for gifting us the wild-type HSV-1 strain. We also thank Chao Huang for help with the artwork. We thank Professor Rongjuan Pei (Wuhan Institute of Virology, Chinese Academy of Sciences) for testing the ALICE system with HCV (GeneBank accession no. AJ242652) and HBV (GeneBank accession no. V01460). We are grateful to all the laboratory members for assistance in this study, especially Chengwei Yi, Jian Jiang, and C. Xu. We also thank the ECNU Multifunctional Platform for Innovation (011) for supporting the mouse studies and the Instruments Sharing Platform of School of Life Sciences, East China Normal University. We also thank the support from the Chinese Academy of Sciences Youth Interdisciplinary Team. This work was financially supported by the grants from the National Natural Science Foundation of China (NSFC: no. 31861143016, no. 31971346, no. 32261160373), the National Key R&D Program of China, Synthetic Biology Research (no. 2019YFA0904500), the Science and Technology Commission of Shanghai Municipality (no. 18JC1411000), and the Fundamental Research Funds for the Central Universities to H.Y. Work conducted at Duke-NUS was partially supported by Singapore National Research Foundation grants (NRF2012NRF-CRP001–056 and NRF2016NRF-NSFC002–013) and National Medical Research Council (OFLCG19May-0034) to L.-F.W. This work was also partially supported by the Young Scientists Fund of the National Natural Science Foundation of China (no. 32201190), China Postdoctoral Science Foundation (no. 2020M681234 and no. BX2021105), and Chongqing Science Function for Post-Doctoral Scientists (no. CSTB2022NSCQ-BHX0034) to Y.W.

## Author contributions

H.Y. and L.-F.W. conceived this study. H.Y., L-F.W., and Y.W. designed the project. Y.W., Y.X., C.T., L.Q., W.C., H.Z., Q.H., Z.D., Z.W., X.W., X.S., C.L., Y.L., S.X., and D.K. performed the experimental work. H.Y., L.-F.W., Y.W., Y.X., C.T., C.L., R.P., D.E.A, F.C., and P.Z. analyzed the results. H.Y., L.-F.W., D.E.A., and Y.W. wrote the manuscript. All authors edited and approved the manuscript.

## Competing interests

The authors declare no competing interests.
