## [Peer Review File · Nature Communications]

Reviewers' Comments:

Reviewer #1:

Remarks to the Author:

This is the third iteration of a study detailing the design and initial “proof-of-concept” testing of a bioengineered cell-based antiviral system termed ALICE purported to “sense-and-destroy” a range of viruses. [redacted], the authors have performed additional experiments to show that ALICE can identify multiple STING-dependent viruses, including DENV-2, SARS-CoV-2, hCoV-229E, HCV, HBV, ADV, and HSV-1 and the modular design is flexible enough to accommodate tandem sgRNAs targeting essential genes of at least two different viruses (ADV and HSV-1) simultaneously. They have also engineered ALICE to produce interferons upon viral sensing as an additional nonspecific immune defense mechanism. The broad-spectrum utility of ALICE was convincingly shown in vitro. The in vivo proof-of-concept was limited to two HSV-1 mouse models that may or may not be translatable. Who (which patients), when (during infection), and where (which form and route) ALICE may be delivered in actual clinical scenarios to have any impact on the course of viral infection is not clear and remains to be determined. The claim that “ALICE systems could combat refractory virus infectious diseases” was never demonstrated.

Reviewer #2:

Remarks to the Author:

The majority concerns have been addressed. However, this reviewer still thinks it is necessary to address the remaining question on the presence and trafficking of ALICE cells in hydrogel with a new experiment.

Manuscript number: NCOMMS-22-26054A-Z—Point-by-point responses to referees' comments:

REVIEWERS' COMMENTS

Reviewer #1 (Remarks to the Author):

This is the third iteration of a study detailing the design and initial “proof-of-concept” testing of a bioengineered cell-based antiviral system termed ALICE purported to “sense-and-destroy” a range of viruses. [Redacted], the authors have performed additional experiments to show that ALICE can identify multiple STING-dependent viruses, including DENV-2, SARS-CoV-2, hCoV-229E, HCV, HBV, ADV, and HSV-1 and the modular design is flexible enough to accommodate tandem sgRNAs targeting essential genes of at least two different viruses (ADV and HSV-1) simultaneously. They have also engineered ALICE to produce interferons upon viral sensing as an additional nonspecific immune defense mechanism. The broad-spectrum utility of ALICE was convincingly shown *in vitro*. The *in vivo* proof-of-concept was limited to two HSV-1 mouse models that may or may not be translatable. Who (which patients), when (during infection), and where (which form and route) ALICE may be delivered in actual clinical scenarios to have any impact on the course of viral infection is not clear and remains to be determined. The claim that “ALICE systems could combat refractory virus infectious diseases” was never demonstrated.

We thank this reviewer's comments.

To further showcase the potential translational applicability of our ALICE technology in clinical scenarios, we have studied the ALICE device designed to sense and destroy HSV-1 in the herpetic simplex keratitis (HSK) mouse model (**Fig. 7**). The HSK mouse model is commonly used in scientific research for evaluating antiviral effects of new antiviral strategies, where mice are infected with HSV-1 following corneal scarification to mimic actual clinical HSV-1 infection scenarios (Yin D, *Nat Biotechnol*, 2021; Yun H, *Immunity*, 2020; Chen WS, *Nat Commun*, 2016).

We have engineered two adeno-associated viral (AAV) vectors encoded with ALICE_{SaCas9+Ab} that simultaneously deliver sgRNA/SaCas9 complex and specific neutralizing antibody to eliminate and block the HSV-1 virus via retrograde transport from corneas to trigeminal ganglia in HSK mouse model. As shown in **Fig. 7**, we have demonstrated that the closed-loop AAV-ALICE_{SaCas9+Ab} could significantly sense and destroy HSV-1 in trigeminal ganglia of HSK mouse model, which could be a promising approach for HSK treatment.

We have turned down the claim that “ALICE systems could combat refractory virus infectious diseases” to “ALICE systems might have the potential to combat refractory virus infectious diseases”.

Reviewer #2 (Remarks to the Author):

The majority concerns have been addressed. However, this reviewer still thinks it is necessary to address the remaining question on the presence and trafficking of ALICE cells in hydrogel with a new experiment.

We sincerely thank for the comments, which provide helpful suggestions for us to perform a study on the performance of hydrogel-based cell therapies. This reviewer wants us to measure the cell viability and cell trafficking in hydrogel. In fact, it has been widely accepted that cells encapsulated in hydrogels can survive well for more than 8 weeks (Yu Y, *Nat Commun*, 2022; Shanbhag S, *Stem Cell Res Ther*, 2021; Li R, *Sci Adv*, 2019). Moreover, our mouse experimental data (Fig. 5) showed that the sense-and-destroy of ALICE cells loaded in the hydrogel could be activated by HSV-1 for 30 days, which can serve as another proof that cells loaded in the hydrogel could survive for a long time.

Additionally, in our experiments, we did not observe that the ALICE cells in the hydrogel traffic to another place in mice. Previous studies have shown that hyaluronic acid (HA)-based hydrogels are hydrophilic, and upon crosslinking of the polymer chains, these gels swell, retaining their three-dimensional (3D) structure without dissolving for at least 6 weeks (Xu Q, *Mater Sci Eng R*, 2021; Yadav A, *J Neuroinflammation*, 2021). Several studies have demonstrated the development of HA-based 3D hydrogels as carriers for seeding and encapsulation of cells for the targeted delivery of therapeutic agents in animals (Amorim S, *Trends Biotechnol*, 2021). According to these existing studies, we do not think it is necessary to repeat these experiments to confirm the presence and trafficking of ALICE cells in hydrogel.

We hope this manuscript will report on the novel viral sense-and-destroy strategy against viral infection, which will set the stage for more in-depth and wider investigations. Indeed, ALICE system delivered by an AAV-vector is a promising strategy to address corneal virus infection diseases, which will be further extensively studied.